# Spontaneous Symmetry Breaking in Generative Diffusion Models

**Gabriel Raya**[1,2]   **Luca Ambrogioni**[3,4]
[1]Jheronimus Academy of Data Science   [2]Tilburg University   [3]Radboud University
[4]Donders Institute for Brain, Cognition and Behaviour
g.raya@jads.nl, l.ambrogioni@donders.ru.nl

## Abstract

Generative diffusion models have recently emerged as a leading approach for generating high-dimensional data. In this paper, we show that the dynamics of these models exhibit a spontaneous symmetry breaking that divides the generative dynamics into two distinct phases: 1) A linear steady-state dynamics around a central fixed-point and 2) an attractor dynamics directed towards the data manifold. These two "phases" are separated by the change in stability of the central fixed-point, with the resulting window of instability being responsible for the diversity of the generated samples. Using both theoretical and empirical evidence, we show that an accurate simulation of the early dynamics does not significantly contribute to the final generation, since early fluctuations are reverted to the central fixed point. To leverage this insight, we propose a Gaussian late initialization scheme, which significantly improves model performance, achieving up to 3x FID improvements on fast samplers, while also increasing sample diversity (e.g., racial composition of generated CelebA images). Our work offers a new way to understand the generative dynamics of diffusion models that has the potential to bring about higher performance and less biased fast-samplers.

## 1 Introduction

In recent years, generative diffusion models (Sohl-Dickstein et al., 2015), also known as score-based diffusion models, have demonstrated significant progress in image (Ho et al., 2020; Song et al., 2021), sound (Chen et al., 2020; Kong et al., 2020; Liu et al., 2023) and video generation (Ho et al., 2022; Singer et al., 2022). These models have not only produced samples of exceptional quality, but also demonstrated a comprehensive coverage of the data distribution. The generated samples exhibit impressive diversity and minimal mode collapse, which are crucial characteristics of high-performing generative models (Salimans et al., 2016; Lucic et al., 2018; Thanh-Tung and Tran, 2020). Diffusion models are defined in terms of a stochastic dynamic that maps a simple, usually Gaussian, distribution into the distribution of the data. In an intuitive sense, the dynamics of a generated sample passes from a phase of equal potentiality, where any (synthetic) datum could be generated, to a denoising phase where the (randomly) "selected" datum is fully denoised. As we shall see in the rest of this paper, this can be interpreted as a form of spontaneous symmetry breaking. As stated by Gross (1996), "The secret of nature is symmetry, but much of the texture of the world is due to mechanisms of symmetry breaking." Surprisingly, the concept of spontaneous symmetry breaking has not yet been examined in the context of generative modeling. This is particularly noteworthy given the importance of spontaneous symmetry breaking in nature, which accounts for the emergence of various phenomena such as crystals (breaking translation invariance), magnetism (breaking rotation invariance), and broken gauge symmetries, a common theme in contemporary theoretical physics.

The concept of spontaneous symmetry breaking is strongly connected with the theory of phase transitions (Stanley, 1971; Donoghue et al., 2014). In high energy physics, experimental evidence of

$$dX_t = -\nabla_x u(X_t, T-t)dt + g(T-t)dW_t$$

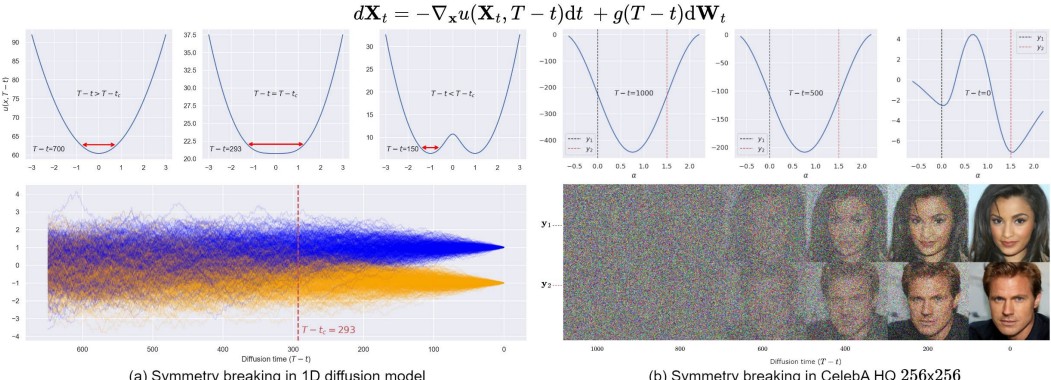

(a) Symmetry breaking in 1D diffusion model      (b) Symmetry breaking in CelebA HQ 256x256

Figure 1: **Overview of spontaneous symmetry breaking in generative diffusion models**.
a) Symmetry breaking in a simple one-dimensional problem with two data points (-1,1). The figures on the top illustrate the potential at different time points, while the bottom figure displays the stochastic trajectories. The red dashed line denotes the time of the spontaneous symmetry breaking (computed analytically). The red arrows represent fluctuations around the fixed-point of the drift. b) Symmetry breaking in a real dataset. The top figures show 1D sections of the potential of a trained diffusion models (CelebA HQ) at different times. The potential is evaluated along circular interpolating paths connecting two generated samples (bottom figure).

particle collisions and decay appears to respect only a subset of the symmetry group that characterizes the theory weak and electromagnetic interactions. This mismatch, which is responsible for the non-null mass of several particles, is thought to be due to the fact that the potential energy of the Higgs field has infinitely many minima, each of which corresponding to a subgroup of the symmetries of the standard model (Englert and Brout, 1964; Higgs, 1964; Anderson, 1963; Nambu and Jona-Lasinio, 1961). Therefore, while the overall symmetry is preserved in the potential, this is not reflected in our physical experiments as we experience a world where only one of these equivalent Higgs state has been arbitrarily "selected". The Higgs potential is often described as a "Mexican hat", since it has an unstable critical point at the origin and a circle of equivalent unstable points around it (see Figure 2). Spontaneous symmetry breaking phenomena are also central to modern statistical physics as they describe the thermodynamics of phase transitions (Stanley, 1971). For example, a ferromagnetic material at low temperature generates a coherent magnetic field in one particular direction since all the atomic dipoles tend to align to the field. On the other hand, at higher temperature the kinetic energy prevents this global alignment and the material does not generate a macroscopic magnetic field. In both cases, the laws of physics are spherically invariant and therefore do not favor any particular directions. However, at low temperature a direction is selected among all the equally likely possibilities, leading to an apparent breaking of the physical symmetries of the system. The global symmetry can then only be recovered by considering an ideal ensemble of many of these magnets, each aligning along one of the equally possible directions.

In this paper, using both theoretical and experimental evidence, we show that the generative dynamics of diffusion models is characterized by a similar spontaneous symmetry breaking phenomenon. In this case, the symmetry group does not come from the laws of physics but it is instead implicit in the dataset. For example, translational invariance can be implicit in the fact that translated versions of similar images are equally represented in a naturalistic dataset. During the early stage of the generative dynamics, each particle reflects all the symmetries since its dynamics fluctuates around a highly symmetric central fixed-point. However, after a critical time, this central fixed-point becomes unstable, and each particle tends towards a different (synthetic) datum with arbitrarily "selected" features, with the global

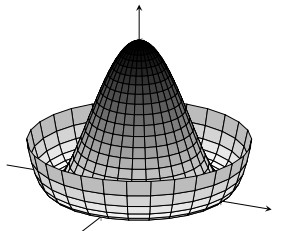

Figure 2: Mexican hat potential

symmetry being apparent only when considering the ensemble of all generated data. An overview of spontaneous symmetry breaking in diffusion models is summarized in Figure 1. Our code can be found at https://github.com/gabrielraya/symmetry_breaking_diffusion_models

## 2 Preliminaries

**Notation**. We denote random variables using upper-case letters and their value using lower-case letters. Additionally, vector-valued variables are denoted using boldface letters. The forward process is denoted as $(\mathbf{Y}_s, s)$, where $\mathbf{Y}_0$ represents the data and $\mathbf{Y}_s$ denotes a noisy state. The generative process is denoted as $(\mathbf{X}_t, t)$, with $s$ and $t = T - s$ representing forward and generative time, respectively, and $T$ as the time horizon. We will always use the standard integration from 0 to $T > 0$ (different to Song et al. (2021)) for the short hand notion of the Itô integral equation. Since we focus on the generative part, we use $\hat{\mathbf{W}}_s$ and $\mathbf{W}_t$ to denote Brownian motion associated to the inference and generative SDEs. For ease of notation, we assume an additive SDE, so $g$ only depends on time.

**Continuous diffusion models**. The stochastic dynamics of a particle $\mathbf{Y}_0 \sim p(\mathbf{y}, 0)$, starting at time $s = 0$, are described as the solution to the Itô SDE: $d\mathbf{Y}_s = f(\mathbf{Y}_s, s)ds + g(s)d\hat{\mathbf{W}}_s$, where $f$ and $g$ are the drift and diffusion coefficient chosen properly such that the marginal density will (approximately) converge to a spherical Gaussian steady-state distribution as $s \to T$. We can express the marginal density at time $s$ as

$$p(\boldsymbol{y}, s) = \int_{\mathbb{R}^D} k(\boldsymbol{y}, s; \boldsymbol{y}_0, 0) p(\boldsymbol{y}_0, 0) d\boldsymbol{y}_0 \, , \tag{1}$$

where $k(\boldsymbol{y}, s; \boldsymbol{y}', s')$ is the transition kernel that 'solves' Eq. 2 below. To generate samples from $p(\boldsymbol{y}, 0)$ by starting from the tractable $p(\boldsymbol{y}, T)$, we can employ a "backward" SDE that reverses this process (Anderson, 1982), whose marginal density evolves according to $p(\boldsymbol{y}, s)$, reverse in time,

$$d\mathbf{X}_t = \left[ g^2(T - t)\nabla_{\boldsymbol{x}} \log p(\mathbf{X}_t, T - t) - f(\mathbf{X}_t, T - t) \right] dt + g(T - t)d\mathbf{W}_t \tag{2}$$

The score function $\nabla_{\boldsymbol{x}} \log p(\boldsymbol{x}, T - t)$ directs the dynamics towards the target distribution $p(\boldsymbol{y}, 0)$ and can be reliably estimated using a denoising autoencoder loss (Vincent, 2011; Song et al., 2021).
**Ornstein–Uhlenbeck process**. In the rest of the paper, we will assume that the forward process follows a (non-stationary) Ornstein–Uhlenbeck dynamics: $d\mathbf{Y}_s = -\frac{1}{2}\beta(s)\mathbf{Y}_s ds + \sqrt{\beta(s)}d\hat{\mathbf{W}}_s$. This is an instance of Variance Preserving (VP) diffusion (Song et al., 2021) wherein the transition kernel can be written in closed form: $k(\boldsymbol{y}, s; \boldsymbol{y}_0, 0) = \mathcal{N}\left(\boldsymbol{y}; \theta_s \boldsymbol{y}_0, (1 - \theta_s^2)I\right)$, with $\theta_s = e^{-\frac{1}{2}\int_0^s \beta(\tau)d\tau}$. It is easy to see that this kernel reduces to an unconditional standard spherical normal for $s \to \infty$ while it tends to a delta function for $s \to 0$.

## 3 Theoretical analysis

For the purpose of our analysis, it is convenient to re-express the generative SDE in Eq. 2 in terms of a potential energy function

$$d\mathbf{X}_t = -\nabla_{\boldsymbol{x}} u(\mathbf{X}_t, T - t)dt + g(T - t)d\mathbf{W}_t \tag{3}$$

where

$$u(\boldsymbol{x}, s) = -g^2(s) \log p(\boldsymbol{x}, s) + \int_{\mathbf{0}}^{\boldsymbol{x}} f(\boldsymbol{z}, s) \cdot d\boldsymbol{z} \, . \tag{4}$$

where the line integral can go along any path connecting $\mathbf{0}$ and $\boldsymbol{x}$. Given a sequence of potential functions $u(\boldsymbol{x}, s)$, we can define an associated symmetry group of transformations

$$G = \{g : \mathbb{R}^D \leftrightarrow \mathbb{R}^D \mid u(g(\boldsymbol{x}), s) = u(\boldsymbol{x}, s), \forall \, s \in \mathbb{R}^+, \boldsymbol{x} \in \mathbb{R}^D\} \, . \tag{5}$$

In words, $G$ is the group of all transformations of the ambient space $R^D$ that preserves the probability measure of the training set at all stages of denoising.

We define a path of fixed points as $\tilde{\boldsymbol{x}}(t) : \mathbb{R} \to \mathbb{R}^D$ such that $\nabla u(\tilde{\boldsymbol{x}}(t), T - t) = 0, \forall t \in \mathbb{R}^+$. These are points of vanishing drift for the stochastic dynamics. The stability of the path can be quantified using the second partial derivatives, which can be organized in the path of Hessian matrices $H(\tilde{\boldsymbol{x}}, T - t)$. A fixed-point is stable when all the eigenvalues of the Hessian matrix of the potential at that point are positive, while it is a saddle or unstable when at least one eigenvalue is negative. Around a stable path of fixed-point, the drift term can be well approximated by a linear function: $\nabla_{\boldsymbol{x}} u(\boldsymbol{x}, T - t) \approx H(\tilde{\boldsymbol{x}}, T - t)(\boldsymbol{x} - \tilde{\boldsymbol{x}})$. From this we can conclude that, along a stable path, the dynamics is locally characterized by the quadratic potential

$$\tilde{u}(\boldsymbol{x}, T - t) = \frac{1}{2}(\boldsymbol{x} - \tilde{\boldsymbol{x}}(t))^T H(\tilde{\boldsymbol{x}}(t), T - t)(\boldsymbol{x} - \tilde{\boldsymbol{x}}(t)) \, . \tag{6}$$

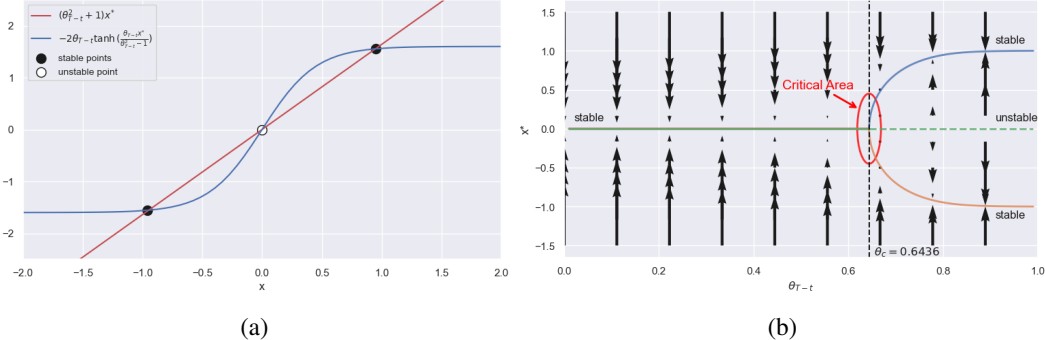

(a)                             (b)

Figure 3: Bifurcation analysis of the generative dynamics of a one-dimensional diffusion model. (a) Geometric visualization of bifurcation of fixed points through the intersection of a straight line and a hyperbolic tangent at a value $\theta_{T-t} > \theta_c$. (b) Bifurcation diagram obtained by numerically solving the self-consistency equation Eq. 10, demonstrating the bifurcation at the critical value $\theta_c$. The blue, orange and green lines denote the three paths of fixed-points. The vector field is given by the drift term (i.e. the gradient of the potential) in the generative SDE.

The associated symmetry group $\tilde{G}$ is generally only a subgroup of the global symmetry group $G$. We say that the dynamics exhibit a *bifurcation* when there are at least two fixed-points paths that overlap for some values of $t$. In this case, usually a stable fixed point loses its stability after a critical time $t_c$ and 'splits' into two or more stable paths. As we will see, this is at the core of the spontaneous symmetry breaking phenomenon. Each of the branched stable paths only preserve a sub-group of the overall symmetry, while the full symmetry is still present when taking all stable paths into account.

### 3.1 Spontaneous symmetry breaking in one-dimensional diffusion models

We start by considering a very simple one-dimensional example with a dataset consisting of two points $y_{-1} = -1$ and $y_1 = -y_{-1} = 1$ sampled with equal probability. In this case, the symmetry group that preserves the potential is comprised by identity and the transformation $g(x) = -x$. Up to terms that are constant in $x$, the potential is given by the following expression:

$$u(x,t) = \beta(T-t)\left(-\frac{1}{4}x^2 - \log\left(e^{-\frac{(x-\theta_{T-t})^2}{2(1-\theta_{T-t}^2)}} + e^{-\frac{(x+\theta_{T-t})^2}{2(1-\theta_{T-t}^2)}}\right)\right) \tag{7}$$

which can be obtained from Eq. 1. Figure 1a illustrates the evolution of the potential (top) and the corresponding one-dimensional generative process (bottom). For all values of $t$, the gradient vanishes at $x = 0$ since the potential is symmetric under the transformation $g(x) = -x$. The stability of this fixed-point can be established by analyzing the second derivative:

$$\left.\frac{\partial^2 u}{\partial x^2}\right|_{x=0} = -\beta(T-t)\left(\frac{1}{2} + \frac{2\theta_{T-t}^2 - 1}{(\theta_{T-t}^2 - 1)^2}\right) \tag{8}$$

where $\theta_{T-t}$ is a monotonic function of $t$ ranging from 0 to 1. The second derivative is positive up to a critical value $\theta_c$ and negative afterwards. This implies that the fixed-point at the origin loses its stability when $\theta_{T-t} > \theta_c$. We can find this critical value by setting the second derivative equal to zero and solving for $\theta_c$, which gives

$$\theta_c = \sqrt{\sqrt{2} - 1} \approx 0.6436 \tag{9}$$

When $\theta_{T-t} > \theta_c$, the origin is not the only fixed-point of the system. All fixed-point can be found by solving the self-consistency equation:

$$(\theta_{T-t}^2 + 1)x^* = -2\theta_{T-t}\tanh\left(\frac{\theta_{T-t}x^*}{\theta_{T-t}^2 - 1}\right) \tag{10}$$

This equation is strikingly similar to the *Curie-Weiss equation of state*, which describes magnetization under the mean-field approximation (Täuber, 2014). Solving this equation corresponds to finding

the intersections between a straight line and an hyperbolic tangent. From Figure 3a, it is clear that there are three solutions for $\theta_{T-t} > \theta_c$ and only the zero solution otherwise. This corresponds to a bifurcation into two paths of fixed points that converge to the values of the data-points for $\theta_{T-t} \to 1$.

We can now describe the spontaneous symmetry breaking phenomenon. The potential $u(x, t)$ is invariant under the transformation $g(x) = -x$ for all values of $\theta_{T-t}$. However, for $\theta_{T-t} > \theta_c$, the individual particles will be 'trapped' in one of those new stable paths of fixed-points, locally breaking the symmetry of the system. From the point of view of generative modeling, this spontaneous symmetry breaking corresponds to the selection of a particular sample among all possible ones. This selection almost exclusively depends on the noise fluctuations around the critical time. In fact, fluctuations for $t \ll t_c$ are irrelevant since the process is mean reverting towards the origin. Similarly, when $t \gg t_c$, fluctuations will be reverted towards the closest fixed-point. However, fluctuations are instead amplified when $t \approx t_c$ since the origin becomes unstable, as illustrated by the red arrows in Figure 1a.

In supplemental section A.2, we provide detailed calculations pertaining to this one-dimensional example. Additionally, in section A.3, we generalize our investigation to models of arbitrarily high dimensionality by considering a hyper-spherical data distribution.

## 3.2 Theoretical analysis of spontaneous symmetry breaking for arbitrary normalized datasets

We can now move to a more realistic scenario where the data is comprised by a finite number $N$ of data-points $\{\boldsymbol{y}_1, \ldots, \boldsymbol{y}_N\}$ embedded in $R^D$. Assuming iid sampling, the most general symmetry group in this case is given by all norm-preserving transformations of $R^D$ that map data-points into data-points. Up to constant terms, the potential is given by

$$u(\boldsymbol{x}, t) = -\beta(t) \left( \frac{1}{4} \|\boldsymbol{x}\|_2^2 + \log \sum_j e^{-\frac{\|\boldsymbol{x} - \theta_{T-t} \boldsymbol{y}_j\|_2^2}{2(1 - \theta_{T-t}^2)}} \right) \tag{11}$$

where the sum runs over the whole dataset. The fixed-points of this model can be found by solving the following self-consistency equation:

$$\frac{1 + \theta_{T-t}^2}{2\theta_{T-t}} \boldsymbol{x}^* = \frac{1}{\sum_j w_j(\boldsymbol{x}^*; \theta_{T-t})} \sum_j w_j(\boldsymbol{x}^*; \theta_{T-t}) \boldsymbol{y}_j \tag{12}$$

where $w_j(\boldsymbol{x}^*; \theta_{T-t}) = e^{-\|\boldsymbol{x}^* - \theta_{T-t} \boldsymbol{y}_j\|_2^2 / (2(1 - \theta_{T-t}^2))}$. While this is a very general case, we can still prove the existence of a spontaneous symmetry breaking at the origin under two mild conditions. First of all, we assume the data-points to be centered: $\sum_j \boldsymbol{y}_j = 0$. Furthermore, we assumed that the data-points are normalized so as to have a norm $r$: $\|\boldsymbol{y}_j\|_2 = r, \forall j$,. Under these conditions, which can be easily enforced on real data through normalization, it is straightforward to see from Eq. 12 that the origin is minimum of the potential for all values of $t$. While we cannot evaluate all the eigenvalues of the Hessian matrix at the origin in closed-form, we can obtain a simple expression for the trace of the Hessian (i.e. the Laplacian of the potential):

$$\nabla^2 u|_{x=0} = -\beta(T - t) \left( \frac{D}{2} + \frac{(D + r^2)\theta_{T-t}^2 - D}{(\theta_{T-t}^2 - 1)^2} \right), \tag{13}$$

which switches sign when $\theta_{T-t}$ is equal to $\theta^* = \sqrt{(\sqrt{D^2 + r^4} - r^2)/D}$. However, in this case we cannot conclude that this is the point of the first spontaneous symmetry breaking since the Laplacian is the sum of all second derivatives, which are not necessarily all equal. Nevertheless, we do know that all second derivatives are positive at the origin for $t \to 0$, since the forward dynamics has a Gaussian steady-state distribution. Therefore, from the change of sign of the Laplacian we can conclude that at least one second derivative at the origin changes sign, corresponding to a change in stability and the onset of a spontaneous symmetry breaking with $\theta^* > \theta_c$. In supplemental section A.4, we provide detailed calculations pertaining to this model.

# 4 Experimental evidence of spontaneous symmetry breaking in trained diffusion models

In this section, we present empirical evidence demonstrating the occurrence of the spontaneous symmetry breaking phenomenon in diffusion models across a range of realistic image datasets, including MNIST, CIFAR-10, CelebA 32x32, Imagenet 64x64 and CelebA 64x64. We trained diffusion models in discrete time (DDPM) with corresponding continuous time Variance-Preserving SDE for each dataset with a time horizon of $T = 1000$ and evaluated fast samplers using Denoising Diffusion Implicit Models (DDIMs) (Song et al., 2020) and Pseudo-numerical Methods for Diffusion Models (PNDM) (Liu et al., 2022). For more detailed information about our experimental implementation, please refer to supplemental material section E.

## 4.1 Analyzing the impact of a late start initialization on FID performance

To investigate how the bifurcation affects generative performance, we performed an exploratory analysis of Fréchet Inception Distance (FID) scores for different diffusion time starts. The objective of this analysis is to examine the possible effects of starting the generative process at a late diffusion time, since our theory predicts that performance will stay approximately constant up to a first critical time and then it will sharply decrease. We explicitly define a "late start" as changing the starting point of the generative process. For instance, instead of starting at $T = 1000$, we evaluate the model's generative performance for various starting points, ranging from $s_{start} = 50$ to $s_{start} = T$, using as starting state $\mathbf{x}_T \sim \mathcal{N}(\mathbf{0}, I)$. In particular, since the distribution is not longer $\mathcal{N}(\mathbf{0}, I)$ for any $s << T$, a decrease in performance can be expected. However, as shown in Figure 4, denoted as "DDPM-1000", there is indeed an initial phase in which the generative performance of the model remains largely unaffected by this distributional mismatch, and in some cases, better FID scores were obtained. Remarkably, as predicted by our framework, in all datasets we see a sharp degradation in performance after a certain initialization time threshold, with an apparent discontinuity in the second derivative of the FID curve. We provide additional results in supplemental section B.1.

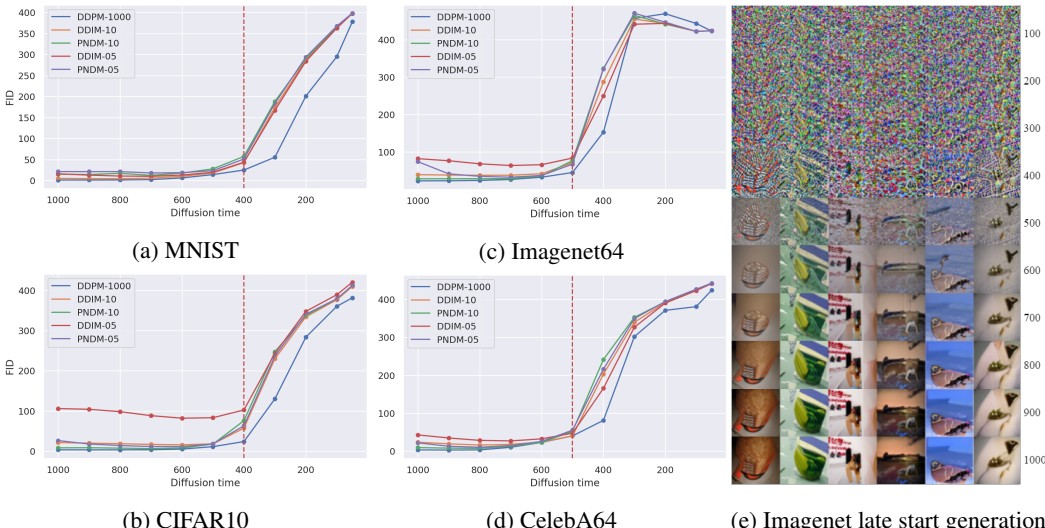

Figure 4: Analysis of the model's performance, as measured by FID scores, for different starting times using three different sampling methods: the normal DDPM sampler with decreasing time steps from $T = 1000$ to 0, and fast sampler DDIM and PSDM for 10 and 5 denoising steps. The vertical line corresponds to the maximum of the second derivative of the FID curve, which offers a rough estimate of the first bifurcation time. (e) Illustrates samples generation on Imagenet64, while progressively varying the starting time from 1000 to 100.

| Dataset | n | gls-DDPM | DDPM |
|---|---|---|---|
| MNIST | 10 | **4.21** | 6.75 |
| | 5 | **6.95** | 13.25 |
| | 3 | **11.92** | 42.63 |
| CIFAR10 | 10 | **28.77** | 43.35 |
| | 5 | **42.46** | 84.82 |
| | 3 | **57.03** | 146.95 |
| CelebA32 | 10 | **11.05** | 26.79 |
| | 5 | **14.79** | 40.92 |
| | 3 | **18.93** | 59.75 |
| Imagenet64 | 10 | **57.31** | 65.68 |
| | 5 | **75.11** | 99.99 |
| | 3 | **91.69** | 145.71 |
| CelebA64 | 10 | **23.79** | 36.66 |
| | 5 | **31.24** | 48.38 |
| | 3 | **37.05** | 62.18 |

(a) DDPM

| Dataset | n | gls-DDIM | DDIM |
|---|---|---|---|
| MNIST | 10 | **2.44** | 4.46 |
| | 5 | **6.95** | 13.25 |
| | 3 | **11.92** | 42.63 |
| CIFAR10 | 10 | **15.98** | 19.79 |
| | 5 | **26.36** | 44.61 |
| | 3 | **42.31** | 109.37 |
| CelebA32 | 10 | **7.27** | 11.37 |
| | 5 | **10.83** | 23.45 |
| | 3 | **16.24** | 45.34 |
| Imagenet64 | 10 | **36.25** | 38.21 |
| | 5 | **52.11** | 68.21 |
| | 3 | **76.92** | 126.3 |
| CelebA64 | 10 | **15.82** | 19.37 |
| | 5 | **22.06** | 28.51 |
| | 3 | **29.96** | 50.304 |

(b) DDIM

| Dataset | n | gls-PNDM | PNDM |
|---|---|---|---|
| MNIST | 10 | **5.02** | 14.36 |
| | 5 | **5.11** | 21.22 |
| | 3 | **38.23** | 154.89 |
| CIFAR10 | 10 | **5.90** | 8.35 |
| | 5 | **9.55** | 13.77 |
| | 3 | **34.20** | 103.11 |
| CelebA32 | 10 | **2.88** | 4.92 |
| | 5 | **4.2** | 6.61 |
| | 3 | **28.60** | 235.87 |
| Imagenet64 | 10 | **27.9** | 28.27 |
| | 5 | **33.35** | 34.86 |
| | 3 | **50.92** | 70.58 |
| CelebA64 | 10 | **6.80** | 8.03 |
| | 5 | **9.26** | 10.26 |
| | 3 | **51.72** | 171.75 |

(c) PNDM

Table 1: Summary of findings regarding image generation quality, as measured by FID scores. The performance of the stochastic DDPM sampler (a) is compared to the deterministic DDIM (b) and PNDM (c) samplers in the vanilla case, as well as our Gaussian late start initialization scheme denoted as "gls". Results are presented for 3, 5, and 10 denoising steps (denoted as "n") across diverse datasets.

## 4.2 Empirical analysis of the potential function in trained diffusion models

The analysis of the FID curves offers indirect evidence of the existence of a critical time that demarcates the generative dynamics of diffusion models trained on real datasets. In order to obtain direct evidence, we need to study the potential associated to trained models. Unfortunately, studying the stability of the initial fixed-point in a trained model is challenging given the high dimensionality and the fact that its location is not available analytically. To reduce the dimensionality, we analyzed how the potential changes across variance-preserving interpolating curves that connect generated images. We defined an interpolating curve as $x(\alpha, t) = \cos(\alpha) x_1(t) + \sin(\alpha) x_2(t)$, where $x_1(t)$ and $x_2(t)$ are two sampled generative paths. Up to constant terms, the potential along the path $\tilde{u}(\alpha, t) = u(x_\alpha, t)$ can be evaluated from the output of the network (i.e. the score) using the fundamental theorem of line integrals (see supplemental section B.2.1). An example of these potential curves is shown in Figure 1b (top), as predicted by the theory, the potential has a convex shape up to a time and then splits into a bimodal shape. A visualization of the average potentials for several datasets is provided in the supplemental section B.2.2. Interestingly, this pairwise splits happens significantly later than the critical transition time observed in the FID analysis. This suggests that, in real datasets, different spontaneous symmetry breaking phenomena happen at different times, with an early one causing a major distributional shift in the overall dynamics (as visible from the FID curves) and later ones splitting the symmetry between individual data-points.

## 5 Improving the performance of fast samplers

In the following, we leverage the spontaneous symmetry breaking phenomenon in order to improve the performance of fast samplers. Since the early dynamics is approximately linear and mean-reverting, the basic idea is to initialize the samplers just before the onset of the instability. This avoids a wasteful utilization of denoising steps in the early phase while also ensuring that the critical window of instability has been appropriately sampled. Unfortunately, while we can prove the existence of a spontaneous symmetry breaking in a large family of models, we can only determine its exact time in highly simplified toy models. We therefore find an "optimized" starting time empirically by generating samples for different starting times (with a fixed number of equally spaced denoising steps) and selecting the one that gives the highest FID scores. From our theoretical analysis it follows that, while the distribution of the particles can drift from $\mathcal{N}(\mathbf{0}, I)$ before the critical time point, it generally remains close to a multivariate Gaussian distribution. Leveraging these insights, we propose a Gaussian initialization scheme to address the distributional mismatch caused by a late start initialization. This scheme involves estimating the mean and covariance matrix of noise-corrupted

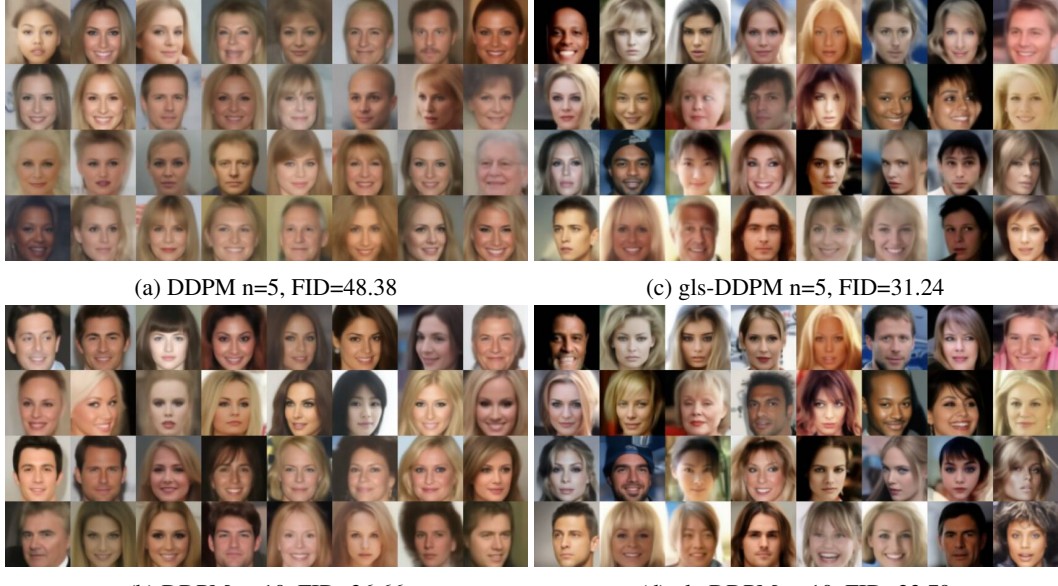

| (a) DDPM n=5, FID=48.38 | (c) gls-DDPM n=5, FID=31.24 |
|---|---|

| (b) DDPM n=10, FID=36.66 | (d) gls-DDPM n=10, FID=23.79 |
|---|---|

Figure 5: Comparison of stochastic DDPM samplers on CelebA64 with varying denoising steps. Subfigures (a) and (c) represent the generative model performance for 5 denoising steps, while (b) and (d) showcase the results for 10 denoising steps. The DDPM was initialized with the common standard initialization point $s_{start} = 800$ for 5 steps and $s_{start} = 900$ for 10 steps. Notably, our Gaussian late start initialization (gls-DDPM) with $s_{start} = 400$ for both 5 and 10 denoising steps demonstrates significant improvements in FID scores and diversity, leveraging spontaneous symmetry breaking in diffusion models.

dataset at the initialization time and utilizing the resulting Gaussian distribution as the initialization point. By using this fitted Gaussian as the starting point, we can obtain an initial sample and run the denoising process from there instead of using a sample from $\mathcal{N}(\mathbf{0}, I)$. We refer to this method as "Gaussian late start" (gls) and present results for stochastic dynamics in Table 1a and deterministic dynamics in Tables 1b and 1c, using DDIM and PNDM samplers respectively. In our experiments on several datasets, we found that in both cases, the Gaussian late start always increases model performance when compared with the baseline samplers. The performance boost is striking in some datasets, with a 2x increase in CelebA 32 for 10 denoising steps and a 3x increase for 5 denoising steps. As evidenced by the results showcased in Figure 5, gls-DDPM boost performance over vanilla DDPM for 10 and 5 denoising steps. For more detailed information on fast samplers, please refer to the supplemental section C. This section also includes extended results for higher number of denoising steps, as detailed in Table 6. Figure 19 provides empirical evidence of the Gaussian nature of the initial distribution via the Shapiro-Wilk test, which remains valid until the critical point.

## 5.1 Diversity analysis

Diffusion models that can quickly generate high-quality samples are highly desirable for practical applications. However, the use of fast samplers can lead to a reduction in the diversity of generated samples. For instance, the analysis shown in Figure 7 using the DDIM sampler with only 5 denoising steps on the CelebA64 dataset revealed a decrease in "race" diversity, with a bias towards the dominant race (white) and a reduction in the coverage of other races. Our theoretical analysis suggests that achieving high generative diversity relies on reliably sampling a narrow temporal window around the critical time, since small perturbations during that window are amplified by the instability and thereby play a central role in the determination of the final samples. This suggests that our Gaussian late initialization should increase diversity when compared with the vanilla fast-sampler, since

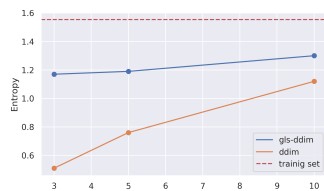

Figure 6: Sample entropy as function of DDIM denoising steps.

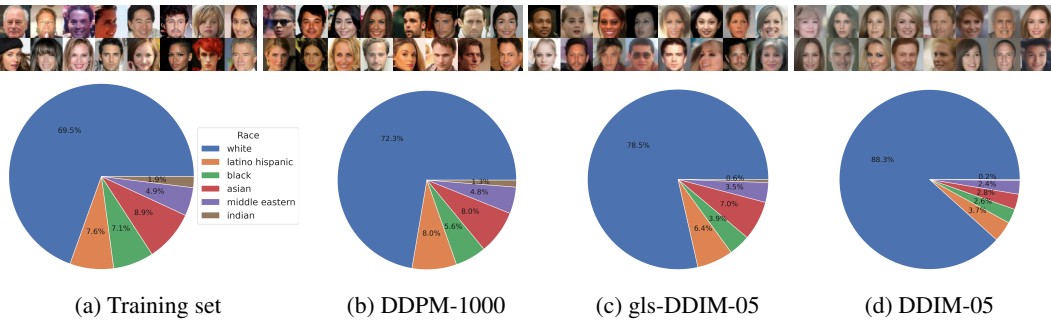

Figure 7: "Race" diversity analysis on CelebA64 over 50,000 generated samples by (c) gls-DDIM and (d) DDIM samplers with 5 denoising steps. Results obtained on (a) training set and (b) DDPM using 1000 denoising steps are provided for reference. Corresponding samples obtained by each set are shown on top of the pie charts.

the Gaussian approximation is reliable prior to the change of shape in the potential. Indeed, our proposed late initialization method (gls-DDIM-05) was able to significantly improve sample diversity, even for a small number of denoising steps, e.g., 3, 5, and 10 sampling steps (see Figure 6). For comparison, we also provided the diversity obtained in the training set and from a typical sampling of DDPM with 1000 sampling steps (DDPM-1000). In supplemental section D, we also provided a full analysis of attributes such as age, gender, race, and emotion by running a facial attribute analysis over 50,000 generated images using the deep face library (Serengil and Ozpinar, 2020).

## 6    Related work

**Efficient sampling in diffusion models**. In recent years, research has been focused on accelerating the slow sampling process in diffusion models, resulting in two categories of efficient samplers: learning-free and learning-based. Among the learning-free samplers, Denoising Diffusion Implicit Models (DDIM) (Song et al., 2020) and Pseudo Numerical Diffusion Models (PNDM) (Liu et al., 2022) have gained considerable attention. DDIM introduced a non-Markovian class of diffusion processes to achieve faster sampling, while PNDM proposed an enhancement to traditional numerical methods through the use of pseudo numerical methods. Both DDIM and PNDM implicitly employ a late start initialization using a linearly spaced open interval running from 1 to $T$, with steps determined by the required number of denoising steps, and consequently not including $T$, but use the sample at $\boldsymbol{x}_T$ as the starting point. The choice of the starting point has been proposed based on empirical results, and to the best of our knowledge, our work is the first to theoretically explore the relevance of this starting point and systematically suggest a way to choose it.

**Diffusion and spontaneous symmetry breaking in optimal control theory**. The concept of spontaneous symmetry breaking has been studied in optimal control theory, with the work of Kappen (2005), who showed the existence of spontaneous symmetry breaking in both stochastic and deterministic optimal control problems. Spontaneous symmetry breaking in the cost as a function of time implies a "delayed choice" mechanism for stochastic control, where taking a decision far in the future is sub-optimal due to high uncertainty. Instead, the optimal policy is to steer between the targets until the critical time one should aim for one of the targets. For $T < t_c$ (far in the past) it is best to steer towards $\boldsymbol{x} = 0$ (between the targets) and delay the choice which slit to aim for until later, when the uncertainty is less close to the terminal state. The relationship between optimal control theory and diffusion models has recently been explored by the work of Berner et al. (2022), who demonstrated that the reverse-backward SDE is directly related to the Hamilton-Jacobi-Bellman equation for the time-inverse log-density through a Hopf-Cole transformation. This theoretical link suggests that the spontaneous symmetry breaking phenomena that we have identified here are analogous to delayed choice phenomena in optimal control.

**Encoding symmetries in generative models**. Symmetries plays a central role in any area of physics, revealing the foundational principles of nature and enabling the formulation of conservation laws (Gross, 1996; Noether, 1971). A substantial body of deep learning literature has capitalized on this inherent property of nature by encoding equivariance to symmetry transformations constrains into deep neural networks (Cohen and Welling, 2016; Worrall and Brostow, 2018; Bogatskiy et al.,

2020; Cesa et al., 2021; Weiler et al., 2021) allowing to construct more efficient and physically interpretable models with minimized learnable parameters. Recently, this attribute has garnered attention in the realm of generative models, spurring interest in integrating both internal (Gauge) and external (space-time) symmetries (Kanwar et al., 2020), particularly when sampling symmetric target densities (Köhler et al., 2020). While the focus has been in incorporating symmetries in normalizing flows (Boyda et al., 2021; Köhler et al., 2020; Rezende et al., 2020; Satorras et al., 2021), there is already an increase interest in encoding symmetries in diffusion models Xu et al. (2022); Hoogeboom et al. (2022). Our work, along with previous research, acknowledges the importance of the existence of symmetry groups for generative modeling. However, the key distinction in our approach lies in emphasizing the relevance of spontaneous symmetry breaking for particle generation rather than encoding symmetry constraints.

**Comparison with other generative models**. While diffusion models differ conceptually from other deep generative approaches like Variational Autoencoders (VAEs) (Kingma and Welling, 2014; Rezende et al., 2014), GANs (Goodfellow et al., 2014), autoregressive models (Van Den Oord et al., 2016), normalizing flows (Dinh et al., 2014; Rezende and Mohamed, 2015), and Energy-Based Models (EBMs), they all share certain operational traits. Specifically, all these models—explicitly or implicitly—start from an isotropic equilibrium state and evolve toward a data-equilibrium state. Beyond this, an underlying theme across these generative models is the idea of pattern formation, marked by a transition from a high-symmetric state to a low-symmetric state. This commonality suggests the potential for leveraging the concept of symmetry breaking for performance improvement. For instance, implementing symmetry breaking mechanisms in GANs could address the well-known problem of mode collapse, thereby enhancing their generative performance. This idea is especially relevant for EBMs, which, like diffusion models, strive to learn an energy function that assigns lower values to states close to the data distribution. Consequently, the phenomenon of spontaneous symmetry breaking could offer key insights for refining the generative capabilities of all these models.

# 7 Discussion and conclusions

In this work, we have demonstrated the occurrence of spontaneous symmetry breaking in diffusion models, a phenomenon of significant importance in understanding their generative dynamics. Our findings challenge the current dominant conception that the generative process of diffusion models is essentially comprised of a single denoising phase. Instead, we have shown that the generative process is divided into two (or more) phases, the early phase where samples are mean-reverted to a fixed point, and the "denoising phase" that moves the particle to the data distribution. Importantly, our analysis suggests that the diversity of the generated samples depends on a point of "critical instability" that divides these two phases. Our experiments demonstrate how this phenomenon can be leveraged to improve the performance of fast samplers using delayed initialization and a Gaussian approximation scheme. However, the algorithm we proposed should be considered as a proof of concept since the full covariance computation does not scale well to high-resolution images. Nevertheless, these limitations can be overcome either by using low-rank initialization or by fitting the Gaussian initialization on a lower-dimensional space of Fourier frequencies. More generally, our work sheds light on the role that symmetries play in the dynamics of deep generative models. Following this research direction, existing theoretical frameworks, such as optimal control and statistical physics, can be leveraged to make advancements in this field towards more efficient, interpretable, and fair models.

## Broader Impact

Our research has important implications for advancing deep generative models, which are essential for understanding complex phenomena like human creativity, molecular formation and protein design. However, the real-world usage of generative diffusion models can pose several risks and reinforce social biases. We hope that the insights and results offered in our work will have a positive social outcome by helping to create less biased models.

# 8 Acknowledgements

We would like to thank Eric Postma for his helpful discussions and comments. Gabriel Raya was funded by the Dutch Research Council (NWO) as part of the CERTIF-AI project (file number 17998).

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
