# Supplementary Material:
# Spontaneous Symmetry Breaking in Generative Diffusion Models

**Gabriel Raya**[1,2]    **Luca Ambrogioni**[3,4]

[1]Jheronimus Academy of Data Science    [2]Tilburg University    [3]Radboud University

[4]Donders Institute for Brain, Cognition and Behaviour

g.raya@jads.nl, l.ambrogioni@donders.ru.nl

This supplementary section aims to provide additional details, derivations, and results that support the main paper. Section A presents detailed mathematical derivations. We start with a brief introduction to diffusion models using a VP-SDE, followed by an explanation of the phenomenon of symmetry breaking in a one-dimensional (Section A.2), hyper-spherical (Section A.3) diffusion model and in normalized datasets (Section A.4). Section B provides additional experiments to support the results reported in the main paper, together with improvements over fast samplers in Section C. A description over results in diversity analysis is given in Section D. Finally, Section E provides a full description of model architectures and detailed experimental settings used to evaluate our experiments.

**Outline**

- Section A : Mathematical derivations

- Section B: Extended experiments

- Section C : Fast samplers results

- Section D: Diversity Analysis

- Section E: Implementation details

## A    Mathematical derivations

### A.1    SDE formulation for analysing symmetry breaking

Assuming $\mathbf{Y}_0$ follows the data distribution $p(y, 0)$ with forward dynamics described by the Îto SDE:

$$d\mathbf{Y}_s = f(\mathbf{Y}_s, s)\mathrm{d}s + g(s)\mathrm{d}\hat{\mathbf{W}}_s \tag{14}$$

and corresponding backward SDE:

$$d\mathbf{X}_t = \Big[g^2(T - t)\nabla_x \log p(\mathbf{X}_t, T - t) - f(\mathbf{X}, T - t)\Big]dt + g(T - t)d\mathbf{W}_t \tag{15}$$

Re-expressing the generative SDE in terms of a potential energy function $u(\mathbf{x}, t)$:

$$u(\mathbf{x}, t) = -g^2(T - t)\log p(\mathbf{x}, T - t) + \int_{\mathbf{0}}^{\mathbf{x}} f(\mathbf{z}, T - t)\mathrm{d}\mathbf{z} \tag{16}$$

yields to the following generative dynamics:

$$d\mathbf{X} = -\nabla_x u(\mathbf{X}_t, T - t)\mathrm{d}t + g(T - t)\mathrm{d}\mathbf{W}_t \tag{17}$$

37th Conference on Neural Information Processing Systems (NeurIPS 2023).

### A.1.1 Variance Preserving SDE as a potential function

We can re-express the widely used Variance Preserving (VP-SDE) (DDPM):

$$d\mathbf{Y}_s = -\frac{1}{2}\beta(s)\mathbf{Y}_s ds + \sqrt{\beta(s)}d\hat{\mathbf{W}}_s \tag{18}$$

with corresponding generative dynamics:

$$d\mathbf{X}_t = \left[\beta(T-t)\nabla_x \log p(\mathbf{X}_t, T-t) + \frac{1}{2}\beta(T-t)\mathbf{X}_t\right]dt + \sqrt{\beta(T-t)}d\mathbf{W}_t \tag{19}$$

in terms of a potential energy $u(\mathbf{X}_t, t)$

$$d\mathbf{X} = -\nabla_x u(\mathbf{X}_t, T-t)dt + \sqrt{\beta(T-t)}\mathbf{W}_t \tag{20}$$

where the potential energy results in the following expression:

$$
\begin{aligned}
u(\mathbf{x}, T-t) &= -\beta(T-t)\log p(\mathbf{x}, T-t) + \int_{\mathbf{0}}^{\mathbf{x}} f(\mathbf{z}, T-t)d\mathbf{z} \\
&= -\beta(T-t)\log p(\mathbf{x}, T-t) - \frac{1}{2}\beta(T-t)\int_{\mathbf{0}}^{\mathbf{x}} \mathbf{Z}_t d\mathbf{z} \\
&= -\beta(T-t)\log p(\mathbf{x}, T-t) - \frac{1}{4}\beta(T-t)\mathbf{X}_t^2
\end{aligned} \tag{21}
$$

with transition kernel expressed in closed form:

$$k(\mathbf{y}, s; \mathbf{y}_0, 0) = \mathcal{N}\left(\mathbf{y}; \theta_s \mathbf{y}_0, (1-\theta_s^2)I\right); \quad \text{where } \theta_s = e^{-\frac{1}{2}\int_0^s \beta(\tau)d\tau}. \tag{22}$$

and where the gradient of the log of the distribution can be reliably estimated using denoising score matching (Song et al., 2021; Ho et al., 2020; Vincent, 2011).

### A.2 Symmetry breaking in one-dimensional diffusion model

We consider a mixture of two delta distributions consisting of two points $x_1 = -x_{-1} = 1$ sampled with equal probability. The distribution at time $s = 0$ is:

$$p(y, 0) = \frac{1}{2}\left(\delta(x+x_1) + \delta(x-x_1)\right)$$

In this case we can compute analytically the marginal distribution $p(y, s)$ as follows:

$$
\begin{aligned}
p(y, s) &= \int k(\mathbf{y}, s; \mathbf{y}_0, 0)p(y_0, 0)dy_0 \\
&= \int \mathcal{N}\left(\mathbf{y}; \theta_s \mathbf{y}_0, (1-\theta_s^2)I\right)\frac{1}{2}\left(\delta(x+x_1) + \delta(x-x_1)\right) \\
&= \frac{1}{2}\int \mathcal{N}\left(\mathbf{y}; \theta_s \mathbf{y}_0, (1-\theta_s^2)I\right)\delta(x-(-x_1))dx \\
&\quad + \frac{1}{2}\int \mathcal{N}\left(\mathbf{y}; \theta_s \mathbf{y}_0, (1-\theta_s^2)I\right)\delta(x-x_1)dx \\
&= \frac{1}{2\sqrt{2\pi(1-\theta_s^2)}}\left(e^{-\frac{(x-\theta_s)^2}{2(1-\theta_s^2)}} + e^{-\frac{(x+\theta_s)^2}{2(1-\theta_s^2)}}\right)
\end{aligned} \tag{23}
$$

Here we used the property of the Direct delta function $\int_{-\infty}^{\infty} f(x)\delta(x-a)dx = f(a)$.
The log probability is expressed as :

$$\log p(y, s) = \log\left(\frac{1}{2\sqrt{2\pi(1-\theta_s^2)}}\left(e^{-\frac{(x-\theta_s)^2}{2(1-\theta_s^2)}} + e^{-\frac{(x+\theta_s)^2}{2(1-\theta_s^2)}}\right)\right) \tag{24}$$

Following Anderson (1982) theorem $\log p(y, s) = \log p(x, t)$ when $s = t$. Therefore, the potential function is given by the following expression :

$$u(x, t) = \beta(T-t)\left(-\frac{1}{4}x^2 - \log\left(e^{-\frac{(x-\theta_{T-t})^2}{2(1-\theta_{T-t}^2)}} + e^{-\frac{(x+\theta_{T-t})^2}{2(1-\theta_{T-t}^2)}}\right)\right) \tag{25}$$

with $s = T - t$.

### A.2.1 Critical point

We now study the stability of the fixed-point at $x = 0$ by analyzing the second derivative. For ease of notation we will use $b = \theta_{T-t}$, $m(x) = \frac{(x-b)^2}{2(b^2-1)}$ and $v(x) = \frac{(x+b)^2}{2(b^2-1)}$.

We first obtain the first derivative of the log term using chain rule:

$$
\begin{aligned}
\frac{\partial}{\partial x} \log \left( e^{m(x)} + e^{v(x)} \right) &= \frac{1}{e^{m(x)} + e^{v(x)}} \cdot \frac{\partial \left( e^{m(x)} + e^{v(x)} \right)}{\partial x} \\
&= \frac{1}{e^{m(x)} + e^{v(x)}} \cdot \left( m(x)' e^{m(x)} + v(x)' e^{v(x)} \right) \\
&= \frac{m(x)' e^{m(x)}}{e^{m(x)} + e^{v(x)}} + \frac{v(x)' e^{v(x)}}{e^{m(x)} + e^{v(x)}}
\end{aligned}
\tag{26}
$$

The second derivative is obtain by deriving each term in the previous results. The derivative of the first RHT is obtained using the quotient rule as follows:

$$
\begin{aligned}
&\frac{\partial}{\partial x} \left( \frac{m(x)' e^{m(x)}}{e^{m(x)} + e^{v(x)}} \right) \\
&= \frac{(m(x)' e^{m(x)})'(e^{m(x)} + e^{v(x)}) - (m(x)' e^{m(x)})(e^{m(x)} + e^{v(x)})'}{(e^{m(x)} + e^{v(x)})^2} \\
&= \frac{(m(x)'' e^{m(x)} + m(x)'^2 e^{m(x)})(e^{m(x)} + e^{v(x)}) - (m(x)' e^{m(x)})(m(x)' e^{m(x)} + v(x)' e^{v(x)})}{(e^{m(x)} + e^{v(x)})^2} \\
&= \frac{m(x)'' e^{m(x)} + m(x)'^2 e^{m(x)}}{e^{m(x)} + e^{v(x)}} - \frac{(m(x)' e^{m(x)})(m(x)' e^{m(x)} + v(x)' e^{v(x)})}{(e^{m(x)} + e^{v(x)})^2}
\end{aligned}
\tag{27}
$$

Similarly, the derivative of the second RHT is obtained as follows:

$$
\begin{aligned}
&\frac{\partial}{\partial x} \left( \frac{v(x)' e^{v(x)}}{e^{m(x)} + e^{v(x)}} \right) \\
&= \frac{(v(x)' e^{v(x)})'(e^{m(x)} + e^{v(x)}) - (v(x)' e^{v(x)})(e^{m(x)} + e^{v(x)})'}{(e^{m(x)} + e^{v(x)})^2} \\
&= \frac{(v(x)'' e^{v(x)} + v(x)'^2 e^{v(x)})(e^{m(x)} + e^{v(x)}) - (v(x)' e^{v(x)})(m(x)' e^{m(x)} + v(x)' e^{v(x)})}{(e^{u(x)} + e^{v(x)})^2} \\
&= \frac{v(x)'' e^{v(x)} + v(x)'^2 e^{v(x)}}{e^{m(x)} + e^{v(x)}} - \frac{(v(x)' e^{v(x)})(m(x)' e^{m(x)} + v(x)' e^{v(x)})}{(e^{m(x)} + e^{v(x)})^2}
\end{aligned}
\tag{28}
$$

with $m(x) = \frac{(x-b)^2}{2(b^2-1)}$, $m(x) = \frac{(x+b)^2}{2(b^2-1)}$, $m(x)' = \frac{(x-b)}{(b^2-1)}$, $v(x)' = \frac{(x+b)}{(b^2-1)}$ and $m(x)'' = \frac{1}{(b^2-1)} = v(x)''$. At $x = 0$, $m(0) = v(0) = \frac{b^2}{2(b^2-1)}$, $m(0)' = -v(0)' = \frac{-b}{(b^2-1)}$ and $m(x)'' = v(x)'' = \frac{1}{(b^2-1)}$. Then, the resulting second derivative is the following:

$$
\begin{aligned}
\frac{\partial^2}{\partial x^2} \log \left( e^{m(x)} + e^{v(x)} \right) &= \frac{\partial}{\partial x} \left( \frac{u(x)' e^{u(x)}}{e^{u(x)} + e^{v(x)}} \right) + \frac{\partial}{\partial x} \left( \frac{v(x)' e^{v(x)}}{e^{u(x)} + e^{v(x)}} \right) \\
&= \frac{u(x)'' e^{u(x)} + u(x)'^2 e^{u(x)}}{e^{u(x)} + e^{v(x)}} - \frac{(u(x)' e^{u(x)})(u(x)' e^{u(x)} + v(x)' e^{v(x)})}{(e^{u(x)} + e^{v(x)})^2} \\
&\quad + \frac{u(x)'' e^{v(x)} + v(x)'^2 e^{v(x)}}{e^{u(x)} + e^{v(x)}} - \frac{(v(x)' e^{v(x)})(u(x)' e^{u(x)} + v(x)' e^{v(x)})}{(e^{u(x)} + e^{v(x)})^2}
\end{aligned}
\tag{29}
$$

at $x = 0$ this becomes :

$$\frac{\partial^2}{\partial x^2} \log\left(e^{m(x)} + e^{v(x)}\right)\Big|_{x=0}$$

$$= \frac{u(0)^{''}e^{u(0)} + u(0)^{'2}e^{u(0)}}{e^{u(0)} + e^{u(0)}} - \frac{(u(0)^{'}e^{u(0x)})(u(0)^{'}e^{u(0)} - u(0)^{'}e^{u(0)})}{(e^{u(0)} + e^{u(0)})^2}$$

$$+ \frac{u(0)^{''}e^{u(0)} + v(0)^{'2}e^{u(0)}}{e^{u(0)} + e^{u(0)}} - \frac{(-u(0)^{'}e^{u(0)})(u(0)^{'}e^{u(0)} - u(0)^{'}e^{u(0)})}{(e^{u(0)} + e^{u(0)})^2}$$

$$= \frac{u(0)^{''}e^{u(0)} + u(0)^{'2}e^{u(0)}}{2e^{u(0)}} + \frac{u(0)^{''}e^{u(0)} + v(0)^{'2}e^{u(0)}}{2e^{u(0)}}$$

$$= \frac{2u(0)^{''} + 2u(0)^{'2}}{2} \quad \text{since } v(0)^{'2} = u(0)^{'2}$$

$$= u(0)^{''} + u(0)^{'2}$$

$$= \frac{1}{b^2 - 1} + \frac{b^2}{(b^2 - 1)^2}$$

$$= \frac{2b^2 - 1}{(b^2 - 1)^2}$$

$$= \frac{2\theta_{T-t}^2 - 1}{(\theta_{T-t}^2 - 1)^2} \qquad \text{by substitution of } b = \theta_{T-t} \tag{30}$$

Consequently the second derivative of the potential is at $x = 0$:

$$\frac{\partial^2 u}{\partial x^2}\Big|_{x=0} = \frac{\partial^2 u}{\partial x^2}\left(-\beta(T-t)\left(\frac{1}{4}x^2 + \log\left(e^{-\frac{(x-\theta_{T-t})^2}{2(1-\theta_{T-t}^2)}} + e^{-\frac{(x+\theta_{T-t})^2}{2(1-\theta_{T-t}^2)}}\right)\right)\right)$$

$$= -\beta(T-t)\left(\frac{1}{2} + \frac{2\theta_{T-t}^2 - 1}{(\theta_{T-t}^2 - 1)^2}\right) \tag{31}$$

The solution to the equation above can be found as follows:

$$0 = -\beta(T-t)\left(\frac{1}{2} + \frac{2\theta_{T-t}^2 - 1}{(\theta_{T-t}^2 - 1)^2}\right)$$

$$= \left(\frac{1}{2}(\theta_{T-t}^2 - 1)^2 + 2\theta_{T-t}^2 - 1\right) \qquad \text{Multiplying two sides by} (\theta_{T-t}^2 - 1)^2$$

$$= \frac{1}{2}\theta_{T-t}^4 - \theta_{T-t}^2 + \frac{1}{2} + 2\theta_{T-t}^2 - 1$$

$$= \frac{1}{2}\left(\theta_{T-t}^4 + 2\theta_{T-t}^2 - 1\right) \tag{32}$$

We can solve this equation using substitution with $x = \theta_{T-t}^2$, resulting in $x^2 + 2x - 1 = (x - 1)$. By the quadratic formula we have:

$$x = \frac{-b \pm \sqrt{b^2 - 4ac}}{2a} = \frac{-2 \pm \sqrt{2^2 + 4}}{2} = \frac{-1 \pm \sqrt{8}}{2} = -1 \pm \sqrt{2} \tag{33}$$

Therefore, $\theta_{T-t}^2 = -1 \pm \sqrt{2}$ with solution:

$$\theta_c = \sqrt{\sqrt{2} - 1} \approx 0.643594$$

Figure 8 illustrates the change of sign at the critical $\theta_c$.

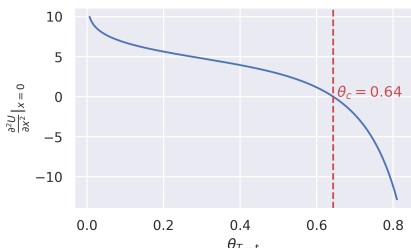

Figure 8: Analysis of the stability of the fixed-point.

### A.2.2 All fixed-points

We now provide the derivation to obtain all the fixed-points at a particular time by analysing its first derivative. For the sake of simplicity, we reorder the terms in the exponential inside the log term, resulting in the following expression:

$$u(x,t) = -\beta(T-t)\left(\frac{1}{4}x^2 + \log\left(e^{\frac{(x-b)^2}{2(b^2-1)}} + e^{\frac{(x+b)^2}{2(b^2-1)}}\right)\right) \tag{34}$$

Again for ease of notation we defined $b = \theta_{T-t}$, and note that we can derive from $\cosh(x) = \frac{e^x + e^{-x}}{2}$ the expression for $2\cosh(x) = e^x + e^{-x}$, which we use to re-express the log term as follows:

$$
\begin{aligned}
\log\left(e^{\frac{(x-b)^2}{2(b^2-1)}} + e^{\frac{(x+b)^2}{2(b^2-1)}}\right) &= \log\left(e^{\frac{x^2-2xb+b^2}{2(b^2-1)}} + e^{\frac{x^2+2xb+b^2}{2(b^2-1)}}\right) \\
&= \log\left(e^{\frac{x^2+b^2}{2(b^2-1)}} \cdot e^{\frac{-xb}{b^2-1}} + e^{\frac{x^2+b^2}{2(b^2-1)}} \cdot e^{\frac{xb}{b^2-1}}\right) \\
&= \log\left(e^{\frac{x^2+b^2}{2(b^2-1)}} \cdot \left(e^{\frac{-xb}{b^2-1}} + e^{\frac{xb}{b^2-1}}\right)\right) \\
&= \log\left(e^{\frac{x^2+b^2}{2(b^2-1)}} \cdot 2\cosh(\frac{xb}{b^2-1})\right) \\
&= \frac{x^2+b^2}{2(b^2-1)} + \log(2) + \log\left(\cosh(\frac{xb}{b^2-1})\right)
\end{aligned}
\tag{35}
$$

Re-expressing the potential, we have:

$$u(x,t) = -\beta(T-t)\left(\frac{1}{4}x^2 + \frac{x^2+b^2}{2(b^2-1)} + \log\left(2\cosh(\frac{xb}{b^2-1})\right)\right) + const \tag{36}$$

computing its derivative we obtain the following

$$\frac{d}{dx}u(x,t) = -\beta(T-t)\left(\frac{1}{2}x + \frac{x}{(b^2-1)} + \frac{b\tanh(\frac{xb}{b^2-1})}{(b^2-1)}\right) \tag{37}$$

Now we solve this equation for $\frac{d}{dx}u(x,t) = 0$

$$\frac{d}{dx}u(x,t) = 0 = -\beta(T-t)\left(\frac{1}{2}x + \frac{x}{(b^2-1)} + \frac{b\tanh(\frac{xb}{b^2-1})}{(b^2-1)}\right)$$

$$-\frac{1}{2}x - \frac{x}{(b^2-1)} = \frac{b\tanh(\frac{xb}{b^2-1})}{(b^2-1)}$$

$$\frac{-x(b^2-1) - 2x}{2(b^2-1)} = \frac{b\tanh(\frac{xb}{b^2-1})}{(b^2-1)}$$

$$\frac{-xb^2 + x - 2x}{2(b^2-1)} = \frac{b\tanh(\frac{xb}{b^2-1})}{(b^2-1)}$$

$$\frac{-xb^2 - x}{2(b^2-1)} = \frac{b\tanh(\frac{xb}{b^2-1})}{(b^2-1)}$$

$$-\frac{x(b^2+1)}{2(b^2-1)} = \frac{b\tanh(\frac{xb}{b^2-1})}{(b^2-1)}$$

$$(b^2+1)x^* = -2b\tanh(\frac{x^*b}{b^2-1}) \tag{38}$$

### A.3 Symmetry breaking in hyper-spherical diffusion models

We will now analyze a more complex multivariate example where the data is sampled from the surface of a $D$-dimensional hyper-sphere. It is easy to see that in this case $G = O(D)$, since both the data distribution and the forward noise are spherically symmetric. Note that, while this is a highly simplified model, it does capture some properties of real data, since the Euclidean norm concentrates in high dimension. In this case, again up to constant terms, the potential is given by:

$$u(\boldsymbol{x},t) = -\beta(T-t)\left(\frac{1}{4}\|\boldsymbol{x}\|_2^2 + \log\left(\int_{R^D} k(\boldsymbol{x}, T-t; \boldsymbol{x}', 0)\phi_D(\boldsymbol{x}'; r)\mathrm{d}\boldsymbol{x}'\right)\right) \tag{39}$$

where $\phi_D(x'; r)$ is a Dirac 'density' spherically symmetric and vanishing outside the surface of the hyper-sphere centered at the origin with radios equal to $r$. Unfortunately, the integral in the potential cannot be solved in closed form. However, it is possible to evaluate its Laplacian (i.e. the trace of the Hessian) at the origin, since the resulting integral only depends on the radial variable. In fact, the Laplacian of our potential at the origin is

$$\nabla^2 u|_{x=0} = -\beta(T-t)\left(\frac{D}{2} + \frac{(D+r^2)\theta_{T-t}^2 - D}{(\theta_{T-t}^2 - 1)^2}\right) \tag{40}$$

In general, the sign of the Laplacian does not contain enough information to determine the stability of the fixed-point. However, in this case the Hessian matrix is a multiple of the identity matrix since all cross-derivatives vanish and all second derivatives have the same value. Consequently, we can determine the critical value of $\theta_{T-t}$ by checking when the Laplacian (and consequently all second derivatives), flips sign. The resulting equation gives us the critical value

$$\theta_c = \sqrt{\frac{\sqrt{D^2 + r^2} - r}{D}} \tag{41}$$

which reduces to Eq. 9 for $r = 1$ and $D = 1$. The qualitative behavior of the hyper-spherical model is analogous to the one-dimensional model. When $\theta_{T-t} < \theta_c$, the origin is the only stable fixed-point. On the other hand, when $\theta_{T-t}$ becomes smaller than $\theta_c$, the origin becomes unstable while it appears a $D-1$-dimensional manifolds of stable points consisting of the surface of a $D$-dimensional sphere centered at the origin with radius equal to $\theta_{T-t}r$. Again, while the potential is spherically symmetric for all values of $\tau$, the symmetry is 'broken' if we consider small perturbations of a single path, since the final position is in an arbitrary location on the surface of the sphere.

## A.4 Symmetry breaking in normalized datasets

### A.4.1 Fixed-points

To derive the fixed-points of a diffusion model with $N$ i.i.d. data-points $\boldsymbol{y}_1, \ldots, \boldsymbol{y}_N \in \mathbb{R}^D$, where the potential is described by Eq. 11, we need to estimate the gradient of the potential function:

To compute the gradient of the potential, first we derive the gradient of the log term:

$$
\frac{\partial}{\partial \boldsymbol{x}}(\log f(\boldsymbol{x})) = \frac{1}{f(\boldsymbol{x})} \frac{\partial}{\partial \boldsymbol{x}} f(\boldsymbol{x}) \qquad \text{with } f(\boldsymbol{x}) = \sum_j e^{-\frac{\|\boldsymbol{x} - \theta_{T-t} \mathbf{y}_j\|_2^2}{2(1 - \theta_{T-t}^2)}}
$$

$$
= \frac{1}{f(\boldsymbol{x})} \sum_j e^{-\frac{\|\boldsymbol{x} - \theta_{T-t} \mathbf{y}_j\|_2^2}{2(1 - \theta_{T-t}^2)}} \frac{\partial}{\partial \boldsymbol{x}}\left(-\frac{\|\boldsymbol{x} - \theta_{T-t} \mathbf{y}_j\|_2^2}{2(1 - \theta_{T-t}^2)}\right) \qquad \text{(chain rule)}
$$

$$
= \frac{1}{f(\boldsymbol{x})}\left(-\sum_j e^{-\frac{\|\boldsymbol{x} - \theta_{T-t} \mathbf{y}_j\|_2^2}{2(1 - \theta_{T-t}^2)}} \frac{\boldsymbol{x} - \theta_{T-t} \mathbf{y}_j}{1 - \theta_{T-t}^2}\right)
$$

$$
= -\frac{1}{\sum_j e^{-\frac{\|\boldsymbol{x} - \theta_{T-t} \mathbf{y}_j\|_2^2}{2(1 - \theta_{T-t}^2)}}} \sum_j e^{-\frac{\|\boldsymbol{x} - \theta_{T-t} \mathbf{y}_j\|_2^2}{2(1 - \theta_{T-t}^2)}} \frac{\boldsymbol{x} - \theta_{T-t} \mathbf{y}_j}{1 - \theta_{T-t}^2} \tag{42}
$$

For ease of notation, we express $b_j = e^{-\frac{\|\boldsymbol{x} - \theta_{T-t} \mathbf{y}_j\|_2^2}{2(1 - \theta_{T-t}^2)}}$, and compute the gradient of the potential:

$$
\frac{\partial}{\partial \boldsymbol{x}} u(\boldsymbol{x}, t) = \frac{\partial}{\partial \boldsymbol{x}}\left(-\beta(T - t)\left(\frac{1}{4}(\|\boldsymbol{x}\|_2^2) + \log \sum_j b_j\right)\right)
$$

$$
= -\beta(T - t)\left(\frac{1}{4} \frac{\partial}{\partial \boldsymbol{x}}(\|\boldsymbol{x}\|_2^2) + \frac{\partial}{\partial \boldsymbol{x}} \log \sum_j b_j\right)
$$

$$
= -\beta(T - t)\left(\frac{1}{2}\boldsymbol{x} - \frac{1}{\sum_j b_j} \sum_j b_j \frac{\boldsymbol{x} - \theta_{T-t} \mathbf{y}_j}{1 - \theta_{T-t}^2}\right) \tag{43}
$$

solve the equation for $\frac{\partial}{\partial \boldsymbol{x}} u(\boldsymbol{x}, t) = 0$

$$\frac{\partial}{\partial \boldsymbol{x}} u(\boldsymbol{x}, t) = 0 = \beta(T - t) \left( -\frac{1}{2}\boldsymbol{x} + \frac{1}{\sum_j b_j} \sum_j b_j \frac{\boldsymbol{x} - \theta_{T-t}\mathbf{y}_j}{1 - \theta_{T-t}^2} \right)$$

$$\frac{1}{2}\boldsymbol{x} = \frac{1}{\sum_j b_j} \sum_j (b_j \frac{\boldsymbol{x} - \theta_{T-t}\mathbf{y}_j}{1 - \theta_{T-t}^2})$$

$$\frac{1}{2}\boldsymbol{x} = \frac{1}{\sum_j b_j} \left( \sum_j \frac{\boldsymbol{x}}{1 - \theta_{T-t}^2} b_j - \sum_j \frac{\theta_{T-t}\boldsymbol{y}_j}{1 - \theta_{T-t}^2} b_j \right)$$

$$\frac{1}{2}\boldsymbol{x} = \frac{\boldsymbol{x}}{1 - \theta_{T-t}^2} \frac{\sum_j b_j}{\sum_j b_j} - \frac{\theta_{T-t}}{1 - \theta_{T-t}^2} \frac{1}{\sum_j b_j} \sum_j b_j \boldsymbol{y}_j$$

$$\frac{1}{2}\boldsymbol{x} - \frac{\boldsymbol{x}}{1 - \theta_{T-t}^2} = -\frac{\theta_{T-t}}{1 - \theta_{T-t}^2} \frac{1}{\sum_j b_j} \sum_j b_j \boldsymbol{y}_j$$

$$\frac{x(1 - \theta_{T-t}^2) - 2\boldsymbol{x}}{2(1 - \theta_{T-t}^2)} = -\frac{\theta_{T-t}}{1 - \theta_{T-t}^2} \frac{1}{\sum_j b_j} \sum_j b_j \boldsymbol{y}_j$$

$$-\boldsymbol{x}\frac{1 + \theta_{T-t}^2}{2(1 - \theta_{T-t}^2)} = -\frac{\theta_{T-t}}{1 - \theta_{T-t}^2} \frac{1}{\sum_j b_j} \sum_j b_j \boldsymbol{y}_j$$

$$\boldsymbol{x}\frac{1 + \theta_{T-t}^2}{2\theta_{T-t}} = \frac{1}{\sum_j b_j} \sum_j b_j \boldsymbol{y}_j \tag{44}$$

Resulting in equation

$$\frac{1 + \theta_{T-t}^2}{2\theta_{T-t}}\boldsymbol{x}^* = \frac{1}{\sum_j e^{w_j(\boldsymbol{x}^*;\theta_{T-t})}} \sum_j e^{w_j(\boldsymbol{x}^*;\theta_{T-t})} \boldsymbol{y}_j \tag{45}$$

### A.4.2 Critical point

We assume data-points to be centered around zero, thus $\sum_j \boldsymbol{y}_j = 0$ , and perturbed samples $\boldsymbol{x} = \{x_1, \ldots, x_D\} \in \mathbb{R}^D$. We denote a single coordinate point $x_i$ at coordinate $i$, then $\sum_i^D 1 = D$.

The Laplacian of the potential function will be calculated as detailed below. A comprehensive derivation of the logarithmic term will subsequently follow:

$$
\begin{aligned}
\nabla^2 u(\boldsymbol{x}, t) &= \sum_i \frac{\partial^2}{\partial x_i^2} u(\boldsymbol{x}, t) \\
&= \sum_i \frac{\partial^2}{\partial^2 x_i} \left( -\beta(T-t)\left( \frac{1}{4}(\|\boldsymbol{x}\|_2^2) + \log \sum_j e^{-\frac{\|\boldsymbol{x} - \theta_{T-t}\mathbf{y}_j\|_2^2}{2(1-\theta_{T-t}^2)}} \right) \right) \\
&= -\beta(T-t)\left( \frac{1}{4}\sum_i \frac{\partial^2}{\partial^2 x_i}(\|\boldsymbol{x}\|_2^2) + \sum_i \frac{\partial^2}{\partial^2 x_i} \log \sum_j e^{-\frac{\|\boldsymbol{x} - \theta_{T-t}\mathbf{y}_j\|_2^2}{2(1-\theta_{T-t}^2)}} \right) \\
&= -\beta(T-t)\left( \frac{D}{2} + \sum_i \frac{\partial^2}{\partial^2 x_i}\left( \log \sum_j e^{-\frac{\|\boldsymbol{x} - \theta_{T-t}\mathbf{y}_j\|_2^2}{2(1-\theta_{T-t}^2)}} \right) \right) \\
&= -\beta(T-t)\left( \frac{D}{2} + \sum_i \frac{\partial^2}{\partial^2 x_i}\left( \log f(\boldsymbol{x}) \right) \right) \\
&= -\beta(T-t)\left( \frac{D}{2} + \sum_i \frac{-1}{1-\theta_{T-t}^2} + \frac{x_i^2}{(1-\theta_{T-t}^2)^2} + \frac{\theta_{T-t}^2}{N(1-\theta_{T-t}^2)^2}\sum_j (\mathbf{y}_i^j)^2 \right) \\
&= -\beta(T-t)\left( \frac{D}{2} + \frac{-D}{1-\theta_{T-t}^2} + \frac{Dx_i^2}{(1-\theta_{T-t}^2)^2} + \frac{\theta_{T-t}^2}{N(1-\theta_{T-t}^2)^2}\sum_j\sum_i (y_i^j)^2 \right) \\
&= -\beta(T-t)\left( \frac{D}{2} + \frac{-D}{1-\theta_{T-t}^2} + \frac{Dx_i^2}{(1-\theta_{T-t}^2)^2} + \frac{\theta_{T-t}^2}{N(1-\theta_{T-t}^2)^2}\sum_j (\mathbf{y}_i^j)^2 \right) \\
&= -\beta(T-t)\left( \frac{D}{2} + \frac{-D}{1-\theta_{T-t}^2} + \frac{Dx_i^2}{(1-\theta_{T-t}^2)^2} + \frac{\theta_{T-t}^2}{N(1-\theta_{T-t}^2)^2}\sum_j r^2 \right) \\
\nabla^2 u(\boldsymbol{x}, t)|_{x=0} &= -\beta(T-t)\left( \frac{D}{2} + \frac{-D}{1-\theta_{T-t}^2} + \frac{\theta_{T-t}^2 r^2}{(1-\theta_{T-t}^2)^2} \right) \\
&= -\beta(T-t)\left( \frac{D}{2} + \frac{-D(1-\theta_{T-t}^2) + \theta_{T-t}^2 r^2}{(1-\theta_{T-t}^2)^2} \right) \\
&= -\beta(T-t)\left( \frac{D}{2} + \frac{(D+r^2)\theta_{T-t}^2 - D}{(1-\theta_{T-t}^2)^2} \right) \\
&= -\beta(T-t)\left( \frac{D}{2} + \frac{(D+r^2)\theta_{T-t}^2 - D}{(\theta_{T-t}^2 - 1)^2} \right)
\end{aligned}
$$

$$(46)$$

where $f(\boldsymbol{x}) = \sum_j e^{-\frac{\|\boldsymbol{x} - \theta_{T-t}\mathbf{y}_j\|_2^2}{2(1-\theta_{T-t}^2)}}$ . Utilizing the quotient rule, we estimated the second partial derivative of the logarithmic term as follows:

$$
\frac{\partial^2}{\partial x^2}(\log f(\boldsymbol{x})) = \frac{\partial}{\partial x}\left( \frac{f'(\boldsymbol{x})}{f(\boldsymbol{x})} \right) = \frac{f''(\boldsymbol{x})f(\boldsymbol{x}) - f'(\boldsymbol{x})^2}{f(\boldsymbol{x})^2} = \frac{f''(\boldsymbol{x})}{f(\boldsymbol{x})^2} \qquad \text{since } f'(\boldsymbol{x})^2|_{x=0} = 0
$$

To compute the second partial derivative we do the following:

1. First compute, we compute the first partial derivative of $f(\boldsymbol{x})$ by applying change rule again

$$
\frac{\partial}{\partial \boldsymbol{x}_i} f(\boldsymbol{x}) = \sum_j \frac{\partial}{\partial x_i} e^{-\frac{\|\boldsymbol{x}-\theta_{T-t}\mathbf{y}_j\|_2^2}{2(1-\theta_{T-t}^2)}}
$$

$$
= \sum_j (-\frac{x_i - \theta_{T-t}\mathbf{y}_i^j}{1 - \theta_{T-t}^2}) e^{-\frac{\|\boldsymbol{x}-\theta_{T-t}\mathbf{y}_j\|_2^2}{2(1-\theta_{T-t}^2)}}
$$

$$
= \sum_j \left(-\frac{1}{1 - \theta_{T-t}^2} x_i + \frac{\theta_{T-t}}{1 - \theta_{T-t}^2}\mathbf{y}_i^j\right) e^{-\frac{\|\boldsymbol{x}-\theta_{T-t}\mathbf{y}_j\|_2^2}{2(1-\theta_{T-t}^2)}} \tag{47}
$$

2. Subsequently, the derivative of $\frac{\partial}{\partial \boldsymbol{x}_i} f(\boldsymbol{x})$ is computed utilizing the product rule:

$$
\frac{\partial^2}{\partial x_i^2} f(\boldsymbol{x}) \tag{48}
$$

$$
= \sum_j \left(-\frac{1}{1 - \theta_{T-t}^2} e^{-\frac{\|\boldsymbol{x}-\theta_{T-t}\mathbf{y}_j\|_2^2}{2(1-\theta_{T-t}^2)}} + (-\frac{x_i - \theta_{T-t}\mathbf{y}_i^j}{1 - \theta_{T-t}^2})^2 e^{-\frac{\|\boldsymbol{x}-\theta_{T-t}\mathbf{y}_j\|_2^2}{2(1-\theta_{T-t}^2)}}\right)
$$

$$
= \sum_j \left(-\frac{1}{1 - \theta_{T-t}^2} + (-\frac{x_i - \theta_{T-t}\mathbf{y}_i^j}{1 - \theta_{T-t}^2})^2\right) e^{-\frac{\|\boldsymbol{x}-\theta_{T-t}\mathbf{y}_j\|_2^2}{2(1-\theta_{T-t}^2)}}
$$

$$
= \sum_j \left(-\frac{1}{1 - \theta_{T-t}^2} + \frac{x_i^2}{(1 - \theta_{T-t}^2)^2} - \frac{2x_i\theta_{T-t}y_i^j}{(1 - \theta_{T-t}^2)^2} + \frac{\theta_{T-t}^2(y_i^j)^2}{(1 - \theta_{T-t}^2)^2}\right) e^{-\frac{\|\boldsymbol{x}-\theta_{T-t}\mathbf{y}_j\|_2^2}{2(1-\theta_{T-t}^2)}}
$$

$$
= \left(\frac{-N}{1 - \theta_{T-t}^2} + \frac{Nx_i^2}{(1 - \theta_{T-t}^2)^2} - \frac{2x_i\theta_{T-t}}{(1 - \theta_{T-t}^2)^2}\underbrace{\sum_j \mathbf{y}_i^j}_{=0} + \frac{\theta_{T-t}^2}{(1 - \theta_{T-t}^2)^2}\sum_j (\mathbf{y}_i^j)^2\right) e^{-\frac{\|\boldsymbol{x}-\theta_{T-t}\mathbf{y}_j\|_2^2}{2(1-\theta_{T-t}^2)}}
$$

$$
= \left(\frac{-N}{1 - \theta_{T-t}^2} + \frac{Nx_i^2}{(1 - \theta_{T-t}^2)^2} + \frac{\theta_{T-t}^2}{(1 - \theta_{T-t}^2)^2}Nr^2\right) e^{-\frac{\|\boldsymbol{x}-\theta_{T-t}\mathbf{y}_j\|_2^2}{2(1-\theta_{T-t}^2)}} \tag{49}
$$

3. We can now proceed to determine the second partial derivative:

$$
\frac{\partial^2}{\partial x_i^2}(\log f(\boldsymbol{x})) = \frac{f''(\boldsymbol{x})}{f(\boldsymbol{x})}
$$

$$
= \frac{\left(\frac{-N}{1-\theta_{T-t}^2} + \frac{Nx_i^2}{(1-\theta_{T-t}^2)^2} + \frac{\theta_{T-t}^2}{(1-\theta_{T-t}^2)^2}\sum_j(\mathbf{y}_i^j)^2\right) e^{-\frac{\|\boldsymbol{x}-\theta_{T-t}\mathbf{y}_j\|_2^2}{2(1-\theta_{T-t}^2)}}}{\sum_j e^{-\frac{\|\boldsymbol{x}-\theta_{T-t}\mathbf{y}_j\|_2^2}{2(1-\theta_{T-t}^2)}}}
$$

$$
= \frac{\left(\frac{-N}{1-\theta_{T-t}^2} + \frac{Nx_i^2}{(1-\theta_{T-t}^2)^2} + \frac{\theta_{T-t}^2}{(1-\theta_{T-t}^2)^2}\sum_j(\mathbf{y}_i^j)^2\right)}{N}
$$

$$
= \frac{-1}{1 - \theta_{T-t}^2} + \frac{x_i^2}{(1 - \theta_{T-t}^2)^2} + \frac{\theta_{T-t}^2}{N(1 - \theta_{T-t}^2)^2}\sum_j (\mathbf{y}_i^j)^2 \tag{50}
$$

### A.4.3 What happens at the origin?

Here, we evaluate Eq. 12 at $\boldsymbol{x}^* = 0$ to assess the behavior at the origin. The initial step involves evaluating the exponential term within the summation:

$$
\begin{aligned}
w_j(\boldsymbol{x}^*; \theta_{T-t})|_{\boldsymbol{x}^*=0} &= e^{-\left\|\boldsymbol{x}^*-\theta_{T-t}\boldsymbol{y}_j\right\|_2^2/(2(1-\theta_{T-t}^2))} = e^{-\left\|\theta_{T-t}\boldsymbol{y}_j\right\|_2^2/(2(1-\theta_{T-t}^2))} \\
&= e^{-\frac{\theta_{T-t}^2}{(2(1-\theta_{T-t}^2))}\left\|\boldsymbol{y}_j\right\|_2^2} \\
&= e^{-\frac{\theta_{T-t}^2}{(2(1-\theta_{T-t}^2))}r^2}
\end{aligned}
\tag{51}
$$

Following this, we calculate the normalized weights at the origin for $N$ data points:

$$
\frac{w_j(\mathbf{0}; \theta_{T-t})}{\sum_j w_j(\mathbf{0}; \theta_{T-t})} = \frac{e^{-\frac{\theta_{T-t}^2}{(2(1-\theta_{T-t}^2))}r^2}}{\sum_j e^{-\frac{\theta_{T-t}^2}{(2(1-\theta_{T-t}^2))}r^2}} = \frac{1}{N}
\tag{52}
$$

Now, we can evaluate Eq. 12 at $\boldsymbol{x}^* = 0$:

$$
\begin{aligned}
\frac{1+\theta_{T-t}^2}{2\theta_{T-t}}\boldsymbol{x}^* &= \frac{1}{\sum_j w_j(\boldsymbol{x}^*; \theta_{T-t})}\sum_j w_j(\boldsymbol{x}^*; \theta_{T-t})\boldsymbol{y}_j \\
\mathbf{0} &= \frac{1}{N}\sum_j \boldsymbol{y}_j \\
\mathbf{0} &= \mathbf{0}
\end{aligned}
\tag{53}
$$

Under the two specific assumptions that 1) data points are centered around zero and 2) constrained to a fixed radius ($r^2 = \left\|\boldsymbol{y}_j\right\|_2^2$), we have demonstrated that the origin indeed functions as a fixed point.

## B  Extended experiments

### B.1  Late start initialization

For completeness, Table 2 presents the resulting Fréchet Inception Distance (FID) scores, as illustrated in Figure 4, for different late start times $s_{\text{start}} \ll T$. Table 3 provides error bars computed from 4 runs. Figure 9 visually demonstrates the unaffected performance of the CIFAR10 model during the early phase, even with fewer denoising steps. Notably, a late start at $s_{\text{start}} = 800$ yields better FID scores compared to $s_{\text{start}} = 1000$, resulting in a direct 20% reduction in compute. Similarly, Figure 10 illustrates generated samples for different late starts using the deterministic DDIM sampler, highlighting how a single starting point remains nearly unaffected in the early generative phase. Figure 11, 12, 13 provide the same analysis but for Celeba64, Imagenet64 and MNIST respectively.

| $s_{start}$ / Dataset | 50 | 100 | 200 | 300 | 400 | 500 | 600 | 700 | 800 | 900 | 1000 |
|---|---|---|---|---|---|---|---|---|---|---|---|
| MNIST | 377.91 | 295.00 | 201.33 | 55.47 | 25.01 | 14.14 | 6.13 | 2.12 | 1.32 | 1.21 | 1.16 |
| CIFAR10 | 381.77 | 360.51 | 283.74 | 130.35 | 23.47 | 11.11 | 5.02 | 3.35 | 3.11 | 3.05 | 3.18 |
| CelebA | 316.6 | 318.04 | 291.57 | 148.23 | 39.85 | 22.51 | 9.76 | 2.76 | 1.91 | 2.06 | 2.15 |
| Imagenet | 423.34 | 443.77 | 469.12 | 456.90 | 152.55 | 45.24 | 32.36 | 26.5 | 23.93 | 22.89 | 22.69 |
| CelebA64 | 424.13 | 380.79 | 371.05 | 301.66 | 81.16 | 40.24 | 23.11 | 10.26 | 3.39 | 3.00 | 3.27 |

Table 2: Analysis of image generation degradation measured in FID scores for different "late start" time $s_{start}$ for DDPM.

| $s_{start}$ / Dataset | 50 | 100 | 200 | 300 | 400 | 500 | 600 | 700 | 800 | 900 | 1000 |
|---|---|---|---|---|---|---|---|---|---|---|---|
| MNIST | 377.83 ± 0.15 | 295.07 ± 0.10 | 201.40 ± 0.27 | 55.64 ± 0.22 | 24.99 ± 0.03 | 13.99 ± 0.26 | 6.20 ± 0.05 | 2.12 ± 0.03 | 1.33 ± 0.01 | 1.22 ± 0.00 | 1.17 ± 0.01 |
| CIFAR10 | 381.79 ± 0.04 | 360.62 ± 0.07 | 283.90 ± 0.14 | 130.15 ± 0.17 | 23.69 ± 0.17 | 11.07 ± 0.05 | 4.98 ± 0.02 | 3.36 ± 0.02 | 3.06 ± 0.01 | 3.06 ± 0.03 | 3.08 ± 0.03 |
| CelebA | 316.59 ± 0.05 | 318.11 ± 0.05 | 291.71 ± 0.09 | 148.40 ± 0.44 | 39.83 ± 0.04 | 22.50 ± 0.08 | 9.83 ± 0.06 | 2.82 ± 0.06 | 1.91 ± 0.01 | 2.08 ± 0.01 | 2.14 ± 0.02 |
| Imagenet | 423.42 ± 0.10 | 443.79 ± 0.04 | 469.37 ± 0.30 | 457.25 ± 0.37 | 152.43 ± 0.47 | 45.12 ± 0.10 | 32.44 ± 0.05 | 26.40 ± 0.09 | 23.92 ± 0.07 | 22.91 ± 0.02 | 22.68 ± 0.03 |

Table 3: Mean FID scores and their associated standard deviations, obtained from a late-start initialization analysis across CIFAR10, MNIST, CelebA, and Imagenet, evaluated over 4 runs.

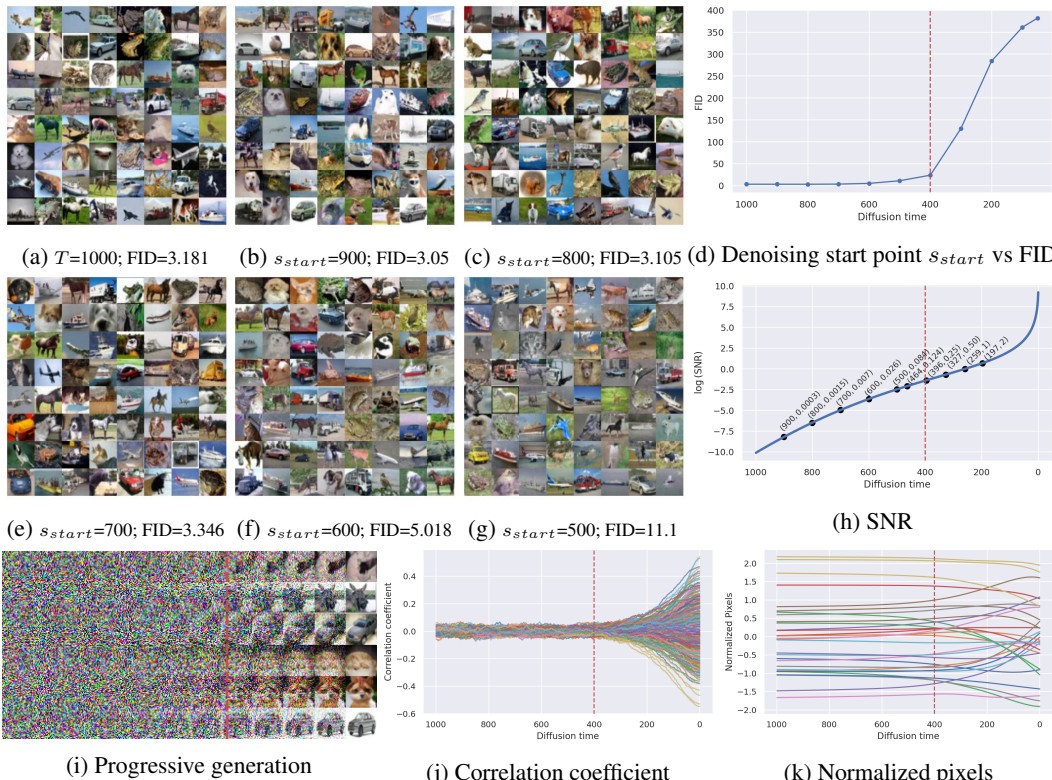

(a) $T$=1000; FID=3.181  (b) $s_{start}$=900; FID=3.05  (c) $s_{start}$=800; FID=3.105  (d) Denoising start point $s_{start}$ vs FID

(e) $s_{start}$=700; FID=3.346  (f) $s_{start}$=600; FID=5.018  (g) $s_{start}$=500; FID=11.1  (h) SNR

(i) Progressive generation  (j) Correlation coefficient  (k) Normalized pixels

Figure 9: Impact of starting the generative process at a late start $s_{start} = 800 << T = 1000$ on the model's performance. **a**) The standard generative process starting at $T = 1000$. **b-g/d**) Depict the process when initiated at $s_{start} << T = 1000$. **d**) Resulting FID scores for different late start times $s_{start} << T$. (i-k) Progressive generation and correlation and normalized pixel analysis (see Sec. B.3).

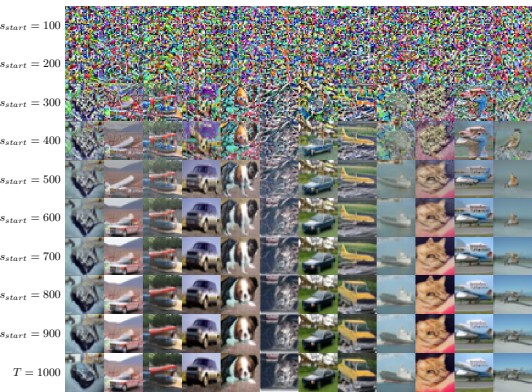

Figure 10: Impact of a late initialization ($s_{start} \ll T = 1000$) on CIFAR10 generation using the DDIM sampler, with starting times varied progressively from 1000 to 100. Despite the late start, the early phase remains largely unaffected since particles convergence to the fixed-point. The number of denoising steps matches each respective starting time, such as 1000 denoising steps for a start at 1000, and 100 for a start at 100.

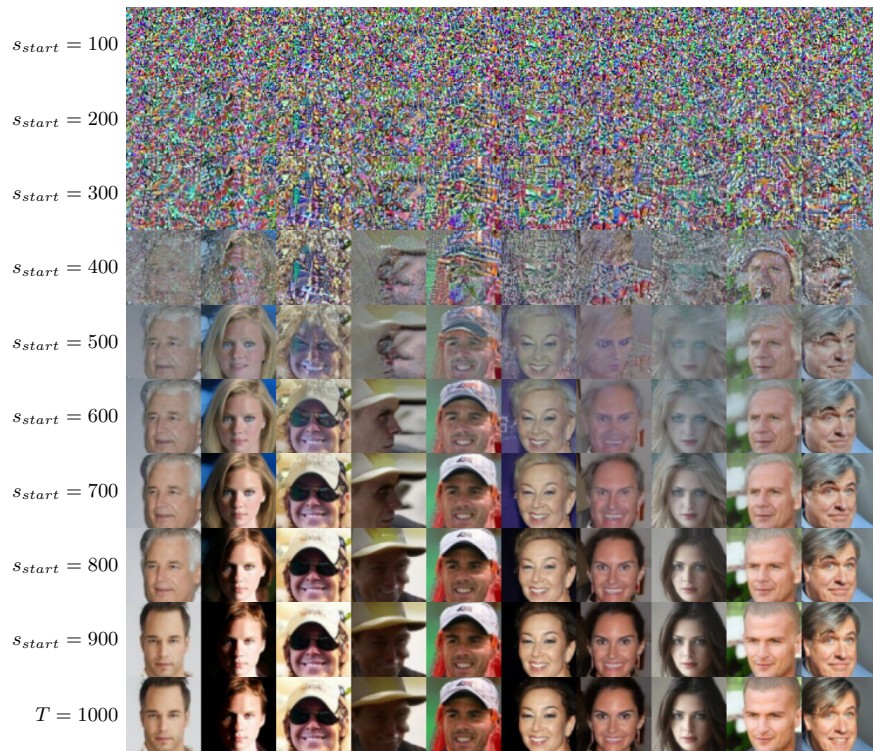

Figure 11: Analogous impact of a late initialization ($s_{\text{start}} \ll T = 1000$) on CelebA64 generation using the DDIM sampler, with starting times varied progressively from 1000 to 100. The denoising steps are set to match each respective starting time, demonstrating similar stability in the early phase.

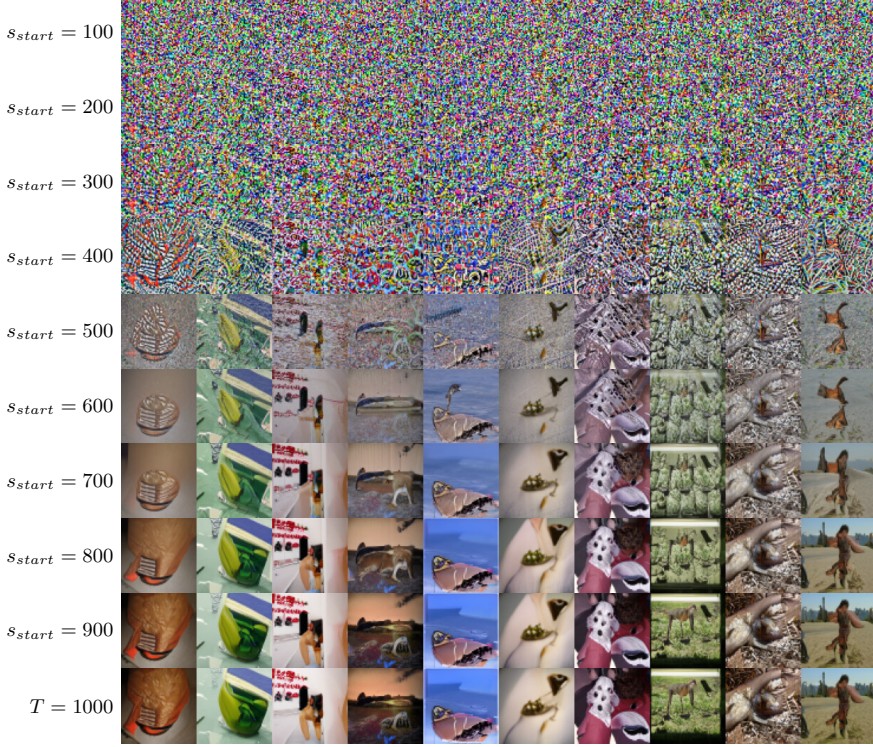

Figure 12: Analogous impact of a late initialization ($s_{\text{start}} \ll T = 1000$) on Imagenet64 generation using the DDIM sampler, with starting times varied progressively from 1000 to 100. The denoising steps are set to match each respective starting time, demonstrating similar stability in the early phase.

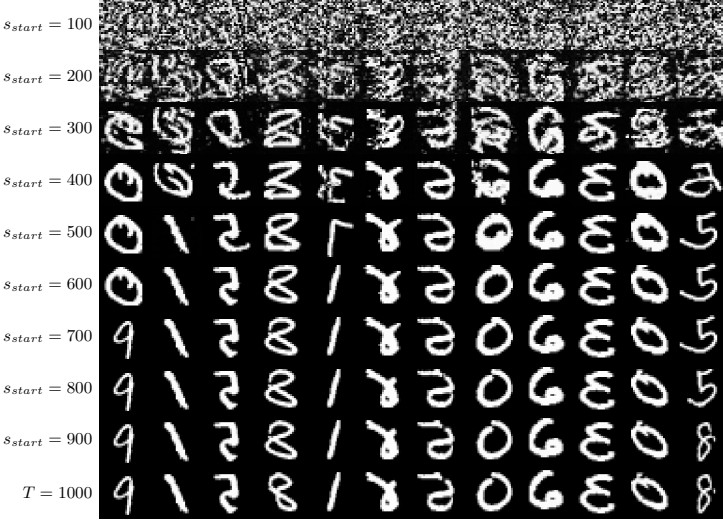

Figure 13: Analogous impact of a late initialization ($s_{\text{start}} \ll T = 1000$) on MNIST generation using the DDIM sampler, with starting times varied progressively from 1000 to 100. The denoising steps are set to match each respective starting time, demonstrating similar stability in the early phase.

## B.2 Empirical analysis of the potential function in trained diffusion models

### B.2.1 Method

To visualize the potential for real high-dimensional datasets, we project the n-dimensional potential onto 1D by focusing only on the trajectory of two sampled generative paths $(\boldsymbol{x}_1(t), \boldsymbol{x}_2(\text{t}))$ and representing the potential as a function of $\alpha$. In short, we do the following:

- We run the sampler and obtain generative trajectory for $\boldsymbol{x}_1(t)$ and for $\boldsymbol{x}_2(t)$, obtaining $\boldsymbol{x}_{1T}, \ldots, \boldsymbol{x}_{1t}, \boldsymbol{x}_{10}$ and $\boldsymbol{x}_{2T}, \ldots, \boldsymbol{x}_{2t}, \boldsymbol{x}_{20}$ corresponding sampled paths.
- We then obtained interpolated curves $\boldsymbol{x}(\alpha, t) = \cos(\alpha)\boldsymbol{x}_1(t) + \sin(\alpha)\boldsymbol{x}_2(t)$ (see Figure 14).
- We then compute the potential (up to a constant) as a function of $\alpha$ by

$$\tilde{u}(\alpha, t) = u(\boldsymbol{x}(\alpha), t) = \int_{-1}^{\alpha} \nabla U(\boldsymbol{x}(a), t) \cdot v \, da$$

in discrete time we estimate

$$\tilde{u}(\alpha_N, t) \approx \sum_{i=1}^{N} \nabla u(x(\alpha_i), t) \cdot v \Delta \alpha$$

where $v = -\sin \alpha \boldsymbol{x}_1 + \cos \alpha \boldsymbol{x}_1$

- Plot potential for $(\alpha, \tilde{u}(\alpha, t))$ coordinates (see Section B.2.2)

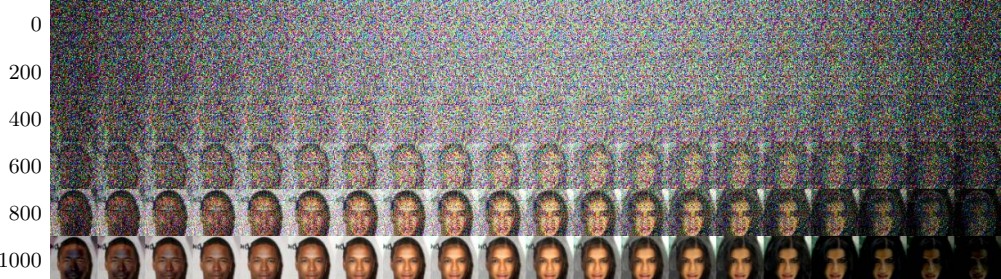

Figure 14: VP interpolations of two sampled paths $\boldsymbol{x}_1(t)$ and $\boldsymbol{x}_2(t)$ over $\alpha \in [-\frac{1}{5}\pi, \frac{7}{10}\pi]$.

### B.2.2 Plots of potentials

This section presents 1D sections plots of the potential of diffusion models trained on CIFAR10, ImageNet64 and CelebA64, along with corresponding generated samples in Figures 15, 16, and 17.

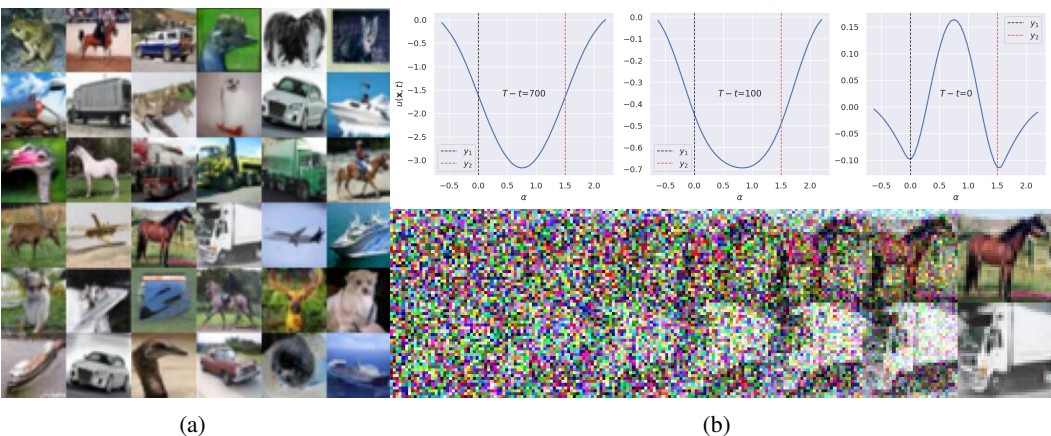

(a)         (b)

Figure 15: Symmetry breaking in CIFAR10: (a) Generated samples; (b) Time-varied 1D potential sections (top figure) from a trained diffusion model along circular paths between two samples (bottom figure), averaged over 20 generated samples from (a).

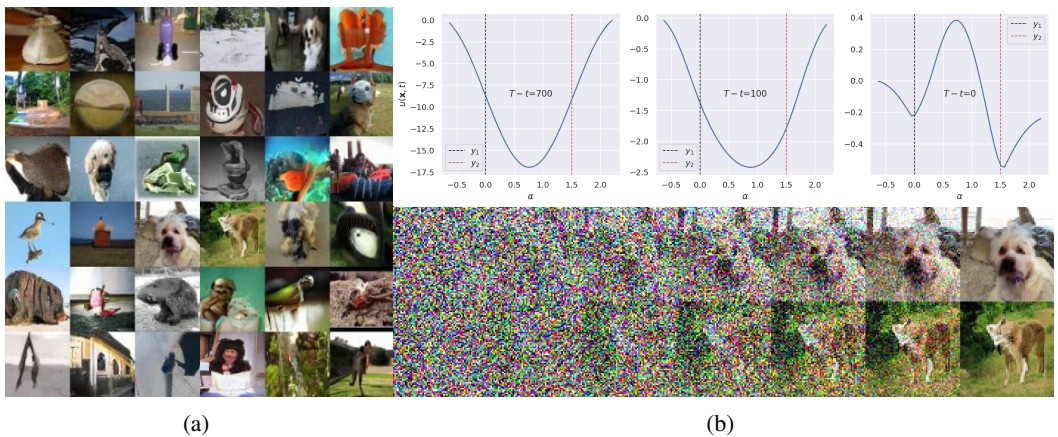

(a)         (b)

Figure 16: Symmetry breaking in Imagenet64.

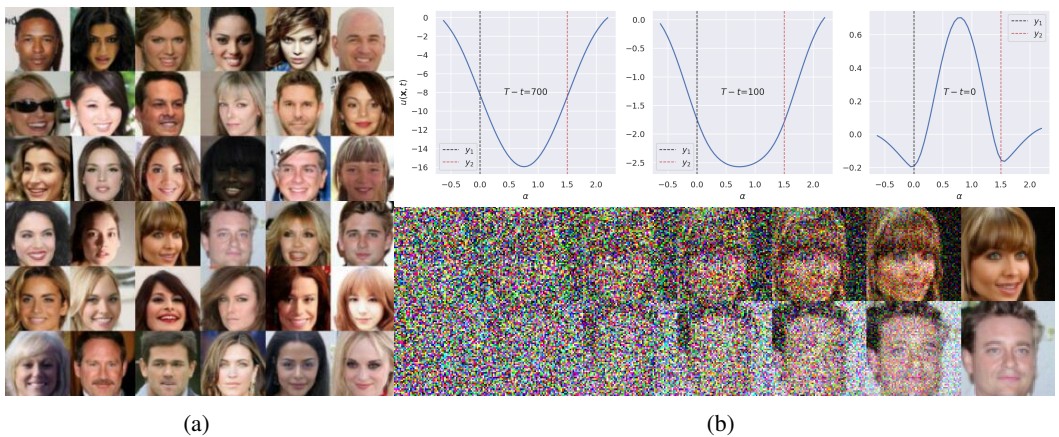

(a)         (b)

Figure 17: Symmetry breaking in CelebA64.

## B.3 Correlation coefficients and Normalized pixel trajectories

In order to qualitatively investigate the generative dynamics of diffusion models, we analyze correlation and normalized pixel trajectories along the generative paths. The correlation coefficient trajectories reveal the evolution of sample correlations over time, highlighting the transition from uncorrelated to aligned samples. By comparing a fixed/reference trajectory with multiple trajectories, we construct correlation trajectories. Furthermore, we track the changes in pixel values using normalized pixel trajectories. Initially, during the early generation phase, the pixel values remain unchanged without any discernible patterns or transformations. However, beyond a critical point, the normalized pixels show the emergence of features and patterns. Our analyses utilize a batch of 300 samples for correlation trajectories and over 500 generated samples for normalized trajectories. The normalized trajectories are sampled using the deterministic DDIM sampler, showcasing both stochastic and deterministic behavior.

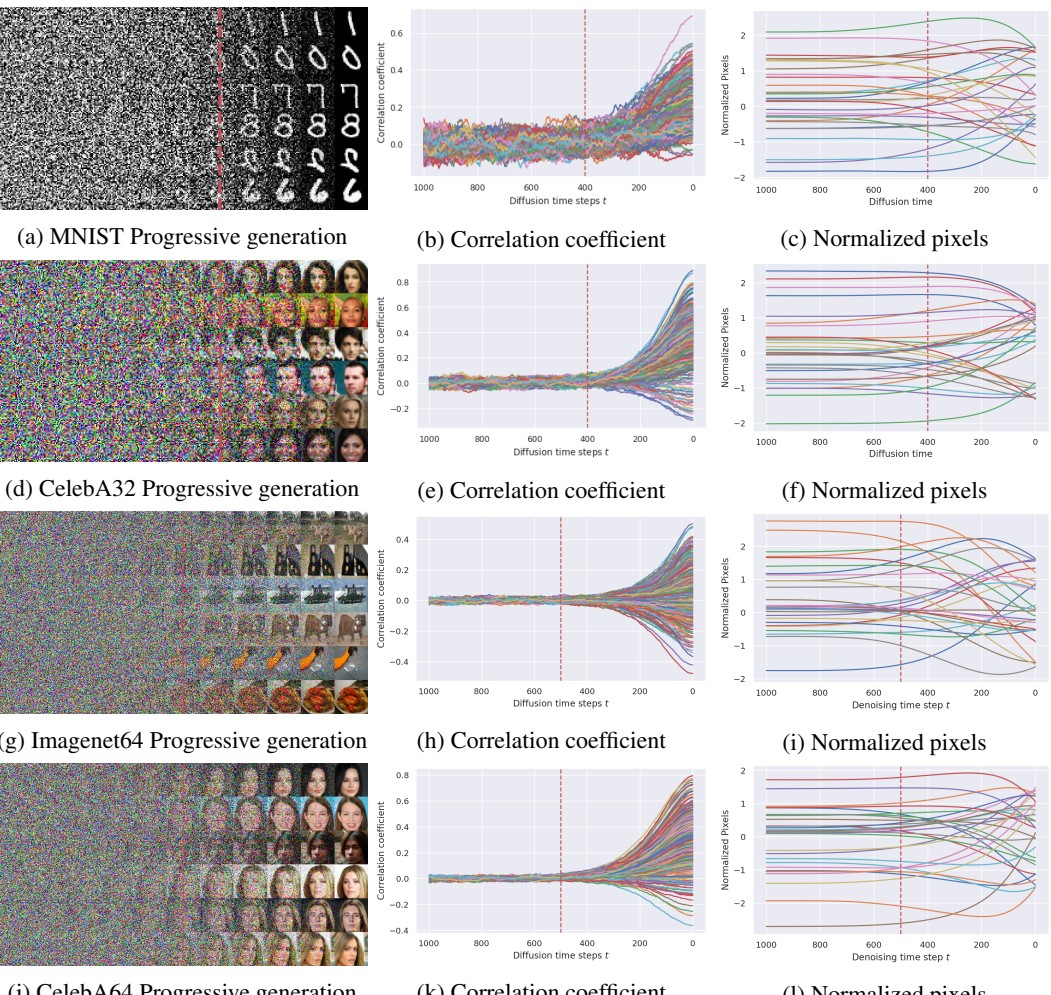

(a) MNIST Progressive generation  (b) Correlation coefficient  (c) Normalized pixels

(d) CelebA32 Progressive generation  (e) Correlation coefficient  (f) Normalized pixels

(g) Imagenet64 Progressive generation  (h) Correlation coefficient  (i) Normalized pixels

(j) CelebA64 Progressive generation  (k) Correlation coefficient  (l) Normalized pixels

Figure 18: Progressive generation and trajectory analysis: Correlation and normalized pixel evolution in diffusion models over time.

# C Fast Samplers

## C.1 Table results

This section provides comprehensive empirical analysis on model performance for varied start times, comparing the Gaussian initialization with vanilla samplers. Table 4 details results for deterministic samplers like DDIM and PSNDM, while Table 5 focuses on stochastic DDPM. These tables include FID scores and consider a fixed number of equally spaced denoising steps. Table 6 extends these results for higher numbers of functions evaluations (NFEs), confirming FID improvements at higher NFEs. Figure 19 validates the Gaussianity assumption in the early generative phase.

| Dataset | model \ $s_{start}$ | 50 | 100 | 200 | 300 | 400 | 500 | 600 | 700 | 800 | 900 | 1000 |
|---|---|---|---|---|---|---|---|---|---|---|---|---|
| MNIST | DDIM-10 | 397.7 | 364.05 | 287.79 | 174.14 | 42.65 | 18.06 | 10.01 | 5.81 | **4.40** | 4.46 | 5.03 |
| | DDIM-10† | 163.22 | 163.83 | 104.34 | 28.98 | 6.07 | **2.44** | 2.5 | 3.05 | 3.57 | 4.20 | 4.80 |
| | DDIM-05 | 397.75 | 362.82 | 283.78 | 167.02 | 43.65 | 19.69 | 13.08 | **10.11** | 10.69 | 12.92 | 16.06 |
| | DDIM-05† | 162.92 | 160.62 | 96.96 | 25.73 | 7.28 | **5.24** | 6.16 | 7.80 | 9.92 | 12.47 | 15.77 |
| | DDIM-03 | 397.49 | 359.51 | 271.84 | 151.35 | 50.50 | **29.5** | 30.01 | 37.93 | 52.11 | 73.81 | 102.28 |
| | DDIM-3† | 162.11 | 155.72 | 84.09 | 22.73 | **11.50** | 13.32 | 20.97 | 34.60 | 50.96 | 73.547 | 102.70 |
| | PSDM-10 | 398.45 | 366.40 | 290.22 | 188.32 | 57.63 | 27.80 | 17.83 | **12.40** | 17.15 | 14.36 | 15.00 |
| | PSDM-10† | 165.47 | 167.99 | 113.20 | 35.25 | 9.35 | **5.02** | 5.78 | 7.31 | 14.75 | 13.32 | 14.77 |
| | PSDM-05 | 398.47 | 367.91 | 293.90 | 183.42 | 51.29 | 23.49 | 18.72 | **18.01** | 21.22 | 21.18 | 21.34 |
| | PSDM-05† | 165.68 | 168.53 | 115.43 | 33.77 | 8.70 | **5.11** | 7.28 | 11.89 | 18.68 | 20.33 | 21.54 |
| | PSDM-03 | 398.81 | 367.92 | 288.95 | 207.45 | 123.95 | **110.05** | 119.18 | 154.89 | 220.26 | 273.10 | 381.89 |
| | PSDM-3† | 165.23 | 166.11 | 117.72 | 43.49 | **38.23** | 64.41 | 98.1 | 148.49 | 218.91 | 271.59 | 381.15 |
| CIFAR10 | DDIM-10 | 410.13 | 376.91 | 334.41 | 229.93 | 57.18 | 17.98 | **15.65** | 16.57 | 18.39 | 19.79 | 21.27 |
| | DDIM-10† | 139.41 | 89.97 | 48.30 | 28.66 | 19.36 | **15.98** | 16.14 | 17.03 | 18.50 | 19.54 | 21.12 |
| | DDIM-5 | 409.17 | 375.47 | 330.77 | 210.12 | 51.78 | **29.97** | 33.55 | 38.28 | 44.61 | 51.31 | 56.68 |
| | DDIM-5† | 132.58 | 86.59 | 49.85 | 33.72 | 26.88 | **26.36** | 30.13 | 36.29 | 44.26 | 51.46 | 56.68 |
| | DDIM-03 | 407.79 | 373.01 | 320.84 | 190.75 | 71.77 | **68.01** | 85.93 | 109.37 | 133.53 | 144.55 | 129.76 |
| | DDIM-3† | 125.29 | 84.14 | 54.08 | 42.84 | **42.31** | 51.8 | 72.86 | 102.88 | 131.68 | 144.51 | 128.91 |
| | PSDM-10 | 412.60 | 379.25 | 337.89 | 247.44 | 75.76 | 17.36 | 7.75 | **6.20** | 7.31 | 8.35 | 8.07 |
| | PSDM-10† | 150.39 | 95.77 | 45.97 | 22.09 | 10.81 | 6.72 | **5.90** | 6.24 | 7.41 | 8.23 | 8.07 |
| | PSDM-05 | 412.49 | 379.41 | 340.89 | 236.68 | 63.11 | 17.78 | **11.13** | 11.34 | 13.77 | 17.56 | 26.15 |
| | PSDM-05† | 149.778 | 92.62 | 44.49 | 21.97 | 12.41 | **9.55** | 10.17 | 11.79 | 13.94 | 17.72 | 25.73 |
| | PSDM-03 | 412.76 | 379.27 | 339.22 | 252.22 | 134.77 | 80.94 | **78.20** | 103.11 | 170.68 | 266.32 | 407.46 |
| | PSDM-3† | 149.63 | 96.97 | 53.34 | 37.10 | **34.20** | 44.98 | 65.78 | 99.46 | 169.76 | 265.58 | 407.63 |
| CelebA (32x32) | DDIM-10 | 345.98 | 316.55 | 306.08 | 251.90 | 68.03 | 25.67 | 25.67 | 15.86 | **10.83** | 11.37 | 13.94 |
| | DDIM-10† | 65.45 | 34.5 | 12.80 | 7.89 | **7.27** | 8.57 | 10.00 | 11.81 | 13.38 | 15.32 | 16.81 |
| | DDIM-05 | 344.04 | 316.53 | 303.82 | 237.93 | 55.65 | 28.74 | 22.11 | **20.03** | 23.45 | 28.61 | 33.21 |
| | DDIM-05† | 59.18 | 30.59 | 13.41 | **10.83** | 11.86 | 14.29 | 17.59 | 21.42 | 25.67 | 30.05 | 33.85 |
| | DDIM-03 | 341.43 | 317.05 | 297.70 | 215.48 | 57.31 | 42.72 | **39.73** | 45.34 | 56.31 | 65.41 | 70.65 |
| | DDIM-3† | 53.38 | 28.27 | **16.24** | 16.29 | 19.67 | 25.42 | 33.12 | 44.41 | 56.39 | 65.77 | 70.87 |
| | PSDM-10 | 350.09 | 316.93 | 306.89 | 263.09 | 94.16 | 31.19 | 15.41 | 6.41 | **4.17** | 4.92 | 5.61 |
| | PSDM-10† | 76.28 | 43.35 | 14.13 | 5.53 | 3.16 | **2.88** | 3.34 | 4.29 | 5.07 | 5.81 | 5.93 |
| | PSDM-05 | 350.62 | 316.52 | 308.97 | 259.61 | 82.27 | 29.94 | 16.16 | 7.88 | **6.61** | 8.46 | 11.07 |
| | PSDM-05† | 74.87 | 39.57 | 12.11 | 5.56 | **4.2** | 4.57 | 5.39 | 6.63 | 8.04 | 9.55 | 11.11 |
| | PSDM-03 | 350.06 | 316.85 | 305.95 | 276.48 | 203.66 | **167.61** | 182.22 | 235.87 | 340.24 | 282.92 | 330.85 |
| | PSDM-3† | 75.43 | 47.62 | **28.60** | 31.63 | 45.56 | 72.57 | 127.70 | 214.48 | 331.07 | 282.69 | 331.07 |
| Imagenet (64x64) | DDIM-10 | 424.13 | 422.57 | 443.86 | 455.37 | 286.98 | 71.39 | 41.62 | 37.79 | **37.56** | 38.21 | 38.21 |
| | DDIM-10† | 263.67 | 213.10 | 103.10 | 66.68 | 53.45 | 44.18 | 38.46 | **36.25** | 36.64 | 37.90 | 39.22 |
| | DDIM-05 | 423.88 | 422.51 | 443.88 | 441.66 | 249.88 | 83.73 | 65.87 | 63.74 | **68.21** | 76.41 | 82.09 |
| | DDIM-05† | 260.95 | 207.15 | 104.13 | 73.28 | 60.98 | 53.83 | **52.11** | 56.41 | 65.21 | 75.99 | 82.22 |
| | DDIM-03 | 423.37 | 422.69 | 438.96 | 414.58 | 225.88 | 130.18 | **119.98** | 126.38 | 147.59 | 171.55 | 184.78 |
| | DDIM-3† | 256.70 | 198.52 | 108.13 | 84.31 | **76.92** | 77.31 | 89.05 | 111.34 | 141.40 | 168.82 | 183.88 |
| | PSDM-10 | 424.26 | 422.59 | 442.03 | 465.26 | 323.34 | 76.74 | 35.71 | 28.64 | 28.43 | **28.27** | **28.21** |
| | PSDM-10† | 267.76 | 219.10 | 103.88 | 61.27 | 45.63 | 36.85 | 31.08 | 28.59 | 27.93 | **27.9** | 28.05 |
| | PSDM-05 | 424.18 | 422.35 | 447.05 | 471.08 | 321.94 | 67.40 | 37.13 | **32.39** | 34.86 | 41.34 | 73.96 |
| | PSDM-05† | 267.73 | 218.87 | 98.16 | 59.60 | 46.32 | 38.41 | 34.24 | **33.35** | 34.97 | 42.02 | 73.91 |
| | PSDM-03 | 424.35 | 422.64 | 443.79 | 466.20 | 329.90 | 115.26 | 73.63 | **70.58** | 153.47 | 306.65 | 391.19 |
| | PSDM-3† | 267.52 | 219.09 | 109.04 | 68.48 | 54.77 | **50.92** | 54.42 | 66.64 | 157.63 | 305.77 | 391.28 |
| CelebA (64x64) | DDIM-10 | 441.61 | 424.22 | 391.42 | 339.56 | 202.98 | 40.51 | 23.91 | 17.95 | **16.13** | 19.37 | 23.60 |
| | DDIM-10† | 93.41 | 65.24 | 34.43 | 22.30 | 17.57 | **15.82** | 16.35 | 18.42 | 21.12 | 22.72 | 24.98 |
| | DDIM-05 | 441.51 | 422.97 | 390.12 | 326.51 | 165.61 | 47.46 | 32.47 | **27.08** | 28.51 | 34.76 | 42.82 |
| | DDIM-05† | 86.56 | 59.26 | 34.13 | 25.86 | 22.79 | **22.06** | 24.16 | 27.91 | 32.88 | 37.27 | 44.18 |
| | DDIM-03 | 440.68 | 420.81 | 387.85 | 315.0 | 146.78 | 62.27 | 52.34 | **50.304** | 59.8 | 75.68 | 84.89 |
| | DDIM-03† | 79.69 | 55.19 | 36.54 | 31.28 | **29.96** | 31.86 | 37.62 | 48.10 | 61.99 | 76.42 | 85.00 |
| | PSDM-10 | 442.67 | 426.44 | 393.62 | 352.46 | 241.59 | 52.21 | 22.91 | 12.95 | **7.40** | 8.03 | 9.43 |
| | PSDM-10† | 104.12 | 77.59 | 39.24 | 20.07 | 11.37 | 7.82 | **6.80** | 7.65 | 9.37 | 10.52 | 10.38 |
| | PSDM-05 | 442.53 | 426.75 | 394.26 | 349.31 | 216.03 | 55.25 | 25.80 | 14.79 | **10.26** | 12.92 | 21.91 |
| | PSDM-05† | 103.41 | 72.79 | 34.73 | 19.33 | 12.74 | 9.88 | **9.26** | 10.92 | 13.58 | 16.09 | 21.39 |
| | PSDM-03 | 442.50 | 426.40 | 393.72 | 352.35 | 293.40 | 204.93 | **169.69** | 171.75 | 355.40 | 331.48 | 416.23 |
| | PSDM-03† | 103.08 | 81.84 | 58.93 | 51.96 | **51.72** | 60.66 | 81.53 | 118.69 | 342.83 | 329.88 | 416.63 |

Table 4: Comparison of image generation using deterministic samplers like DDIM and PSNDM, measured in FID Scores. The strategies employed involve different 'late start' scenarios with **5** and **10** denoising steps. PNDM for T starts at 999. †Gaussian approximation initialization (gls).

Table 5

| Dataset | model | 50 | 100 | 200 | 300 | 400 | 500 | 600 | 700 | 800 | 900 | 999 |
|---|---|---|---|---|---|---|---|---|---|---|---|---|
| MNIST | DDPM-10 | 381.52 | 302.46 | 211.31 | 58.27 | 25.73 | 16.61 | 9.37 | **6.24** | 6.25 | 6.75 | 7.34 |
| | DDPM-10† | 164.89 | 159.73 | 66.46 | 12.28 | 4.62 | **4.21** | 4.82 | 5.58 | 6.17 | 6.71 | 7.25 |
| | DDPM-05 | 386.63 | 314.07 | 226.71 | 67.14 | 28.16 | 19.37 | 13.27 | **11.24** | 13.25 | 16.02 | 18.78 |
| | DDPM-05† | 162.03 | 155.05 | 64.62 | 13.27 | **6.95** | 7.26 | 8.82 | 10.63 | 13.12 | 15.98 | 18.88 |
| | DDPM-03 | 391.53 | 330.81 | 238.59 | 86.89 | 36.93 | **29.30** | 32.90 | 42.63 | 54.93 | 73.87 | 100.78 |
| | DDPM-3† | 158.89 | 148.52 | 63.22 | 16.44 | **11.92** | 15.29 | 26.19 | 41.16 | 54.75 | 73.68 | 100.75 |
| CIFAR10 | DDPM-10 | 383.98 | 361.29 | 279.54 | 95.56 | 37.04 | 38.50 | 36.16 | **36.09** | 39.2 | 43.35 | 47.76 |
| | DDPM-10† | 112.94 | 72.63 | 44.39 | 33.60 | 29.10 | **28.77** | 31.13 | 35.05 | 39.32 | 43.63 | 47.80 |
| | DDPM-05 | 388.03 | 362.67 | 281.87 | 90.79 | **56.40** | 65.41 | 68.15 | 74.56 | 84.82 | 96.54 | 105.49 |
| | DDPM-05† | 111.3 | 76.52 | 52.59 | 44.85 | **42.46** | 46.46 | 57.10 | 70.46 | 84.4 | 96.40 | 106.05 |
| | DDPM-03 | 393.92 | 364.69 | 289.34 | 103.07 | **85.80** | 108.30 | 125.24 | 146.95 | 172.63 | 192.61 | 190.69 |
| | DDPM-3† | 112.34 | 80.34 | 60.91 | **57.03** | 60.85 | 76.23 | 104.00 | 139.32 | 171.82 | 193.07 | 191.66 |
| CelebA (32x32) | DDPM-10 | 317.33 | 319.39 | 290.02 | 116.35 | 34.40 | 27.02 | 20.94 | **20.41** | 23.95 | 26.79 | 28.90 |
| | DDPM-10† | 44.88 | 21.05 | **11.05** | 11.30 | 13.59 | 16.37 | 19.13 | 22.49 | 24.99 | 27.24 | 29.31 |
| | DDPM-05 | 319.6 | 319.17 | 289.0 | 109.2 | 40.80 | 36.60 | **32.67** | 34.93 | 40.92 | 46.06 | 50.35 |
| | DDPM-05† | 42.56 | 21.86 | **14.79** | 16.41 | 20.05 | 24.56 | 29.92 | 36.23 | 41.66 | 46.32 | 50.24 |
| | DDPM-03 | 324.14 | 318.42 | 288.56 | 118.83 | 51.77 | **51.34** | 51.12 | 59.75 | 71.29 | 78.1 | 82.21 |
| | DDPM-3† | 42.29 | 23.91 | **18.93** | 22.06 | 27.38 | 35.049 | 44.9 | 57.95 | 70.84 | 78.13 | 82.09 |
| Imagenet (64x64) | DDPM-10 | 422.90 | 439.20 | 456.48 | 418.56 | 121.52 | 66.33 | 63.22 | **60.66** | 62.69 | 65.68 | 69.25 |
| | DDPM-10† | 247.69 | 175.06 | 86.68 | 73.9 | 66.62 | 59.70 | **57.31** | 58.67 | 61.79 | 65.81 | 69.27 |
| | DDPM-05 | 422.63 | 434.66 | 445.04 | 394.01 | 125.68 | 95.92 | 95.45 | **95.06** | 99.99 | 108.84 | 115.235 |
| | DDPM-05† | 246.76 | 174.24 | 95.91 | 85.59 | 79.61 | **75.11** | 77.26 | 85.94 | 96.88 | 108.33 | 115.94 |
| | DDPM-03 | 422.68 | 429.65 | 433.43 | 373.03 | 153.07 | **133.66** | 139.13 | 145.71 | 160.1 | 178.46 | 177.59 |
| | DDPM-3† | 245.64 | 176.23 | 105.28 | 94.93 | **91.69** | 93.7 | 106.34 | 128.62 | 153.00 | 177.08 | 177.43 |
| CelebA (64x64) | DDPM-10 | 428.98 | 383.53 | 367.83 | 301.06 | 62.62 | 42.01 | 32.05 | **27.76** | 30.80 | 36.66 | 40.49 |
| | DDPM-10† | 73.35 | 46.52 | 28.73 | 24.91 | **23.79** | 25.12 | 27.95 | 32.24 | 35.89 | 38.93 | 41.04 |
| | DDPM-05 | 433.17 | 390.28 | 368.18 | 305.34 | 71.96 | 55.1 | 45.32 | **41.90** | 48.38 | 55.42 | 61.94 |
| | DDPM-05† | 68.60 | 45.71 | 33.93 | 31.58 | **31.24** | 33.62 | 38.08 | 44.67 | 51.67 | 56.75 | 62.33 |
| | DDPM-03 | 435.8 | 402.04 | 371.31 | 312.56 | 85.84 | 71.33 | 63.4 | **62.18** | 74.22 | 88.36 | 92.14 |
| | DDPM-3† | 67.18 | 47.58 | 38.32 | **37.05** | 37.85 | 41.49 | 48.71 | 60.55 | 75.79 | 88.55 | 92.25 |

Table 5: Stochastic sampler. †Gaussian approximation initialization (gls).

| Dataset | n | gls-DDPM | DDPM | gls-DDIM | DDIM | gls-PNDM | PNDM |
|---|---|---|---|---|---|---|---|
| MNIST | 20 | **2.40** | 4.10 | **1.21** | 2.39 | **3.70** | 5.39 |
| | 50 | **1.29** | 2.15 | **0.84** | 1.56 | **3.69** | 4.78 |
| | 100 | **0.98** | 1.51 | **0.78** | 1.44 | **3.69** | 4.65 |
| CIFAR10 | 20 | **19.78** | 26.13 | **10.92** | 12.42 | **4.27** | 5.13 |
| | 50 | **12.54** | 16.257 | **7.54** | 8.26 | **3.42** | 3.60 |
| | 100 | **9.16** | 11.55 | **6.19** | 6.57 | **3.32** | 3.34 |
| CelebA64 | 20 | **18.37** | 28.40 | **11.78** | 15.44 | **4.63** | 6.10 |
| | 50 | **12.61** | 20.23 | **7.43** | 10.78 | **3.57** | 4.06 |
| | 100 | **9.27** | 15.33 | **5.51** | 8.13 | **3.40** | 3.58 |

Table 6: FID score comparison for higher NFEs with $n = 20, 50, 100$ denoising steps. Stochastic DDPM, deterministic DDIM, and PNDM samplers are evaluated in both vanilla and "gls" settings.

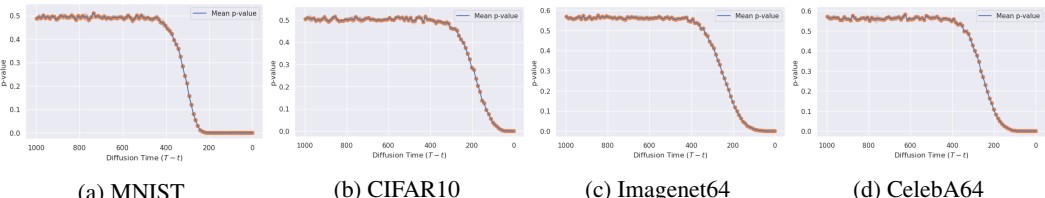

(a) MNIST  (b) CIFAR10  (c) Imagenet64  (d) CelebA64

Figure 19: The Shapiro-Wilk test assesses the normality of data over time, evaluated over 500 perturbed samples. It helps determine if the data closely follows a Multivariate Gaussian distribution up to a specific critical time.

## C.2 Generated images over improved fast samplers

This section presents the enhanced performance of standard samplers due to our Gaussian late start (gls) initialization, visualized across several datasets. Figure 20 displays results of DDIM and PSDM samplers on CelebA64 with five denoising steps. Additional DDIM results are provided in Figure 21 for 5 and 10 denoising steps. The method's performance on MNIST, CIFAR10, ImageNet64, and CelebA64 is further illustrated in Figures 22, 23, 24, and 25 respectively.

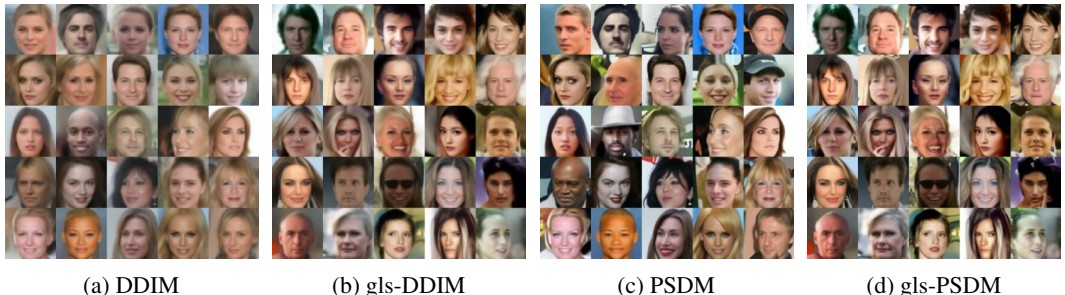

    (a) DDIM          (b) gls-DDIM          (c) PSDM          (d) gls-PSDM

Figure 20: Comparison of deterministic samplers with and without our proposed Gaussian late start for CelebA 64x64 for 5 denoising steps generation.

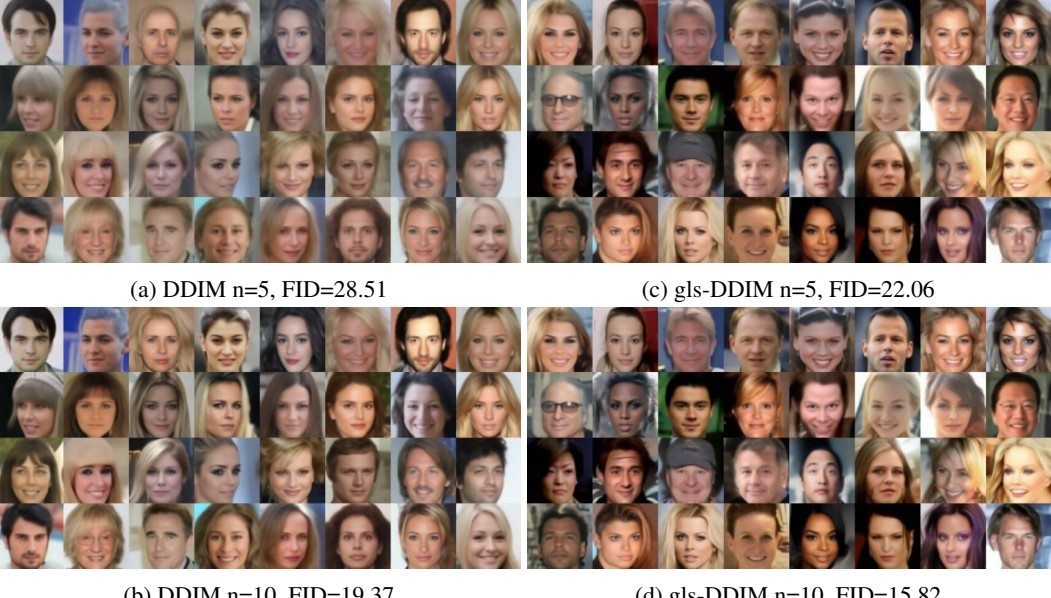

    (a) DDIM n=5, FID=28.51             (c) gls-DDIM n=5, FID=22.06

    (b) DDIM n=10, FID=19.37             (d) gls-DDIM n=10, FID=15.82

Figure 21: Comparison of the deterministic DDIM sampler on CelebA 64x64 with varying denoising steps. Subfigures (a) and (c) represent the generative model performance for 5 denoising steps, while (b) and (d) showcase the results for 10 denoising steps. The DDIM sampler was initialized with the common standard initialization point $s_{start} = 800$ for 5 steps and $s_{start} = 900$ for 10 steps. Notably, our Gaussian late start initialization (gls-DDPM) with $s_{start} = 500$ for both 5 and 10 denoising steps demonstrates significant improvements in FID scores and diversity, leveraging spontaneous symmetry breaking in diffusion models.

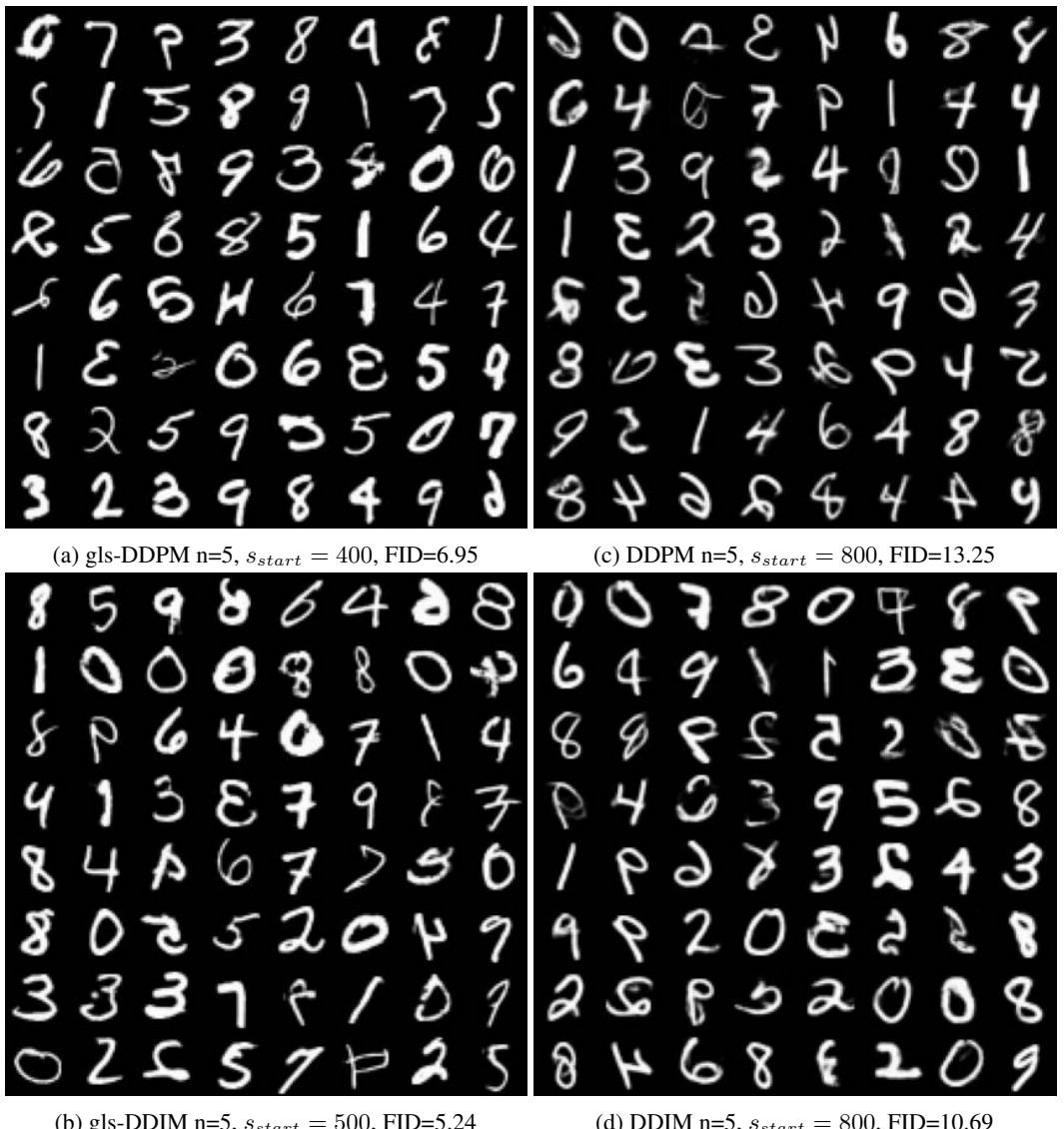

(a) gls-DDPM n=5, $s_{start} = 400$, FID=6.95

(c) DDPM n=5, $s_{start} = 800$, FID=13.25

(b) gls-DDIM n=5, $s_{start} = 500$, FID=5.24

(d) DDIM n=5, $s_{start} = 800$, FID=10.69

Figure 22: Our Gaussian late start initialization boost performance on fast sampler, expemplified here in both DDPM (top row) and DDIM (bottom row) for 5 denoising steps on MNIST.

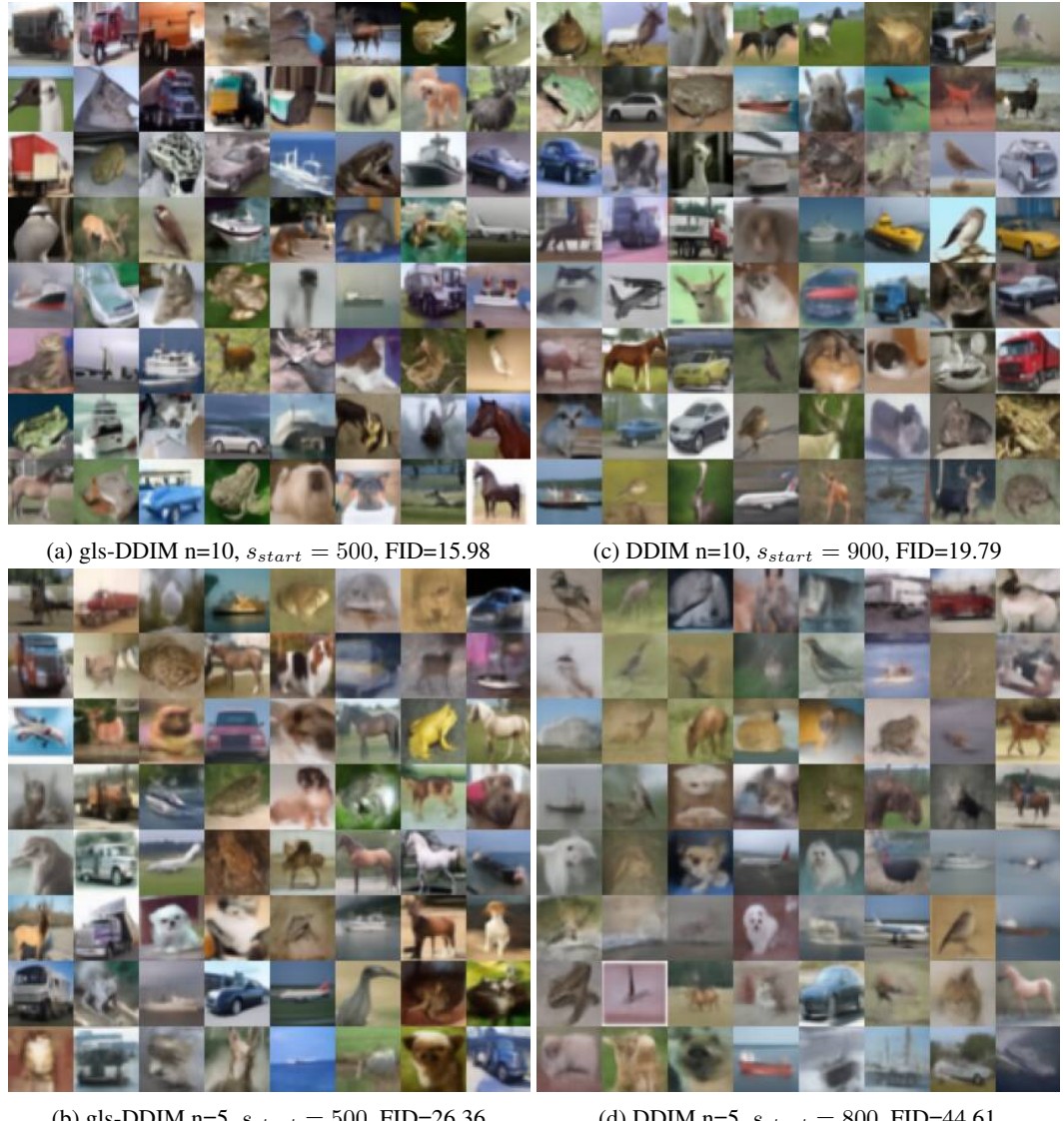

(a) gls-DDIM n=10, $s_{start} = 500$, FID=15.98

(c) DDIM n=10, $s_{start} = 900$, FID=19.79

(b) gls-DDIM n=5, $s_{start} = 500$, FID=26.36

(d) DDIM n=5, $s_{start} = 800$, FID=44.61

Figure 23: Our Gaussian late start initialization boost performance on DDIM for 10 and 5 denoising steps on CIFAR10.

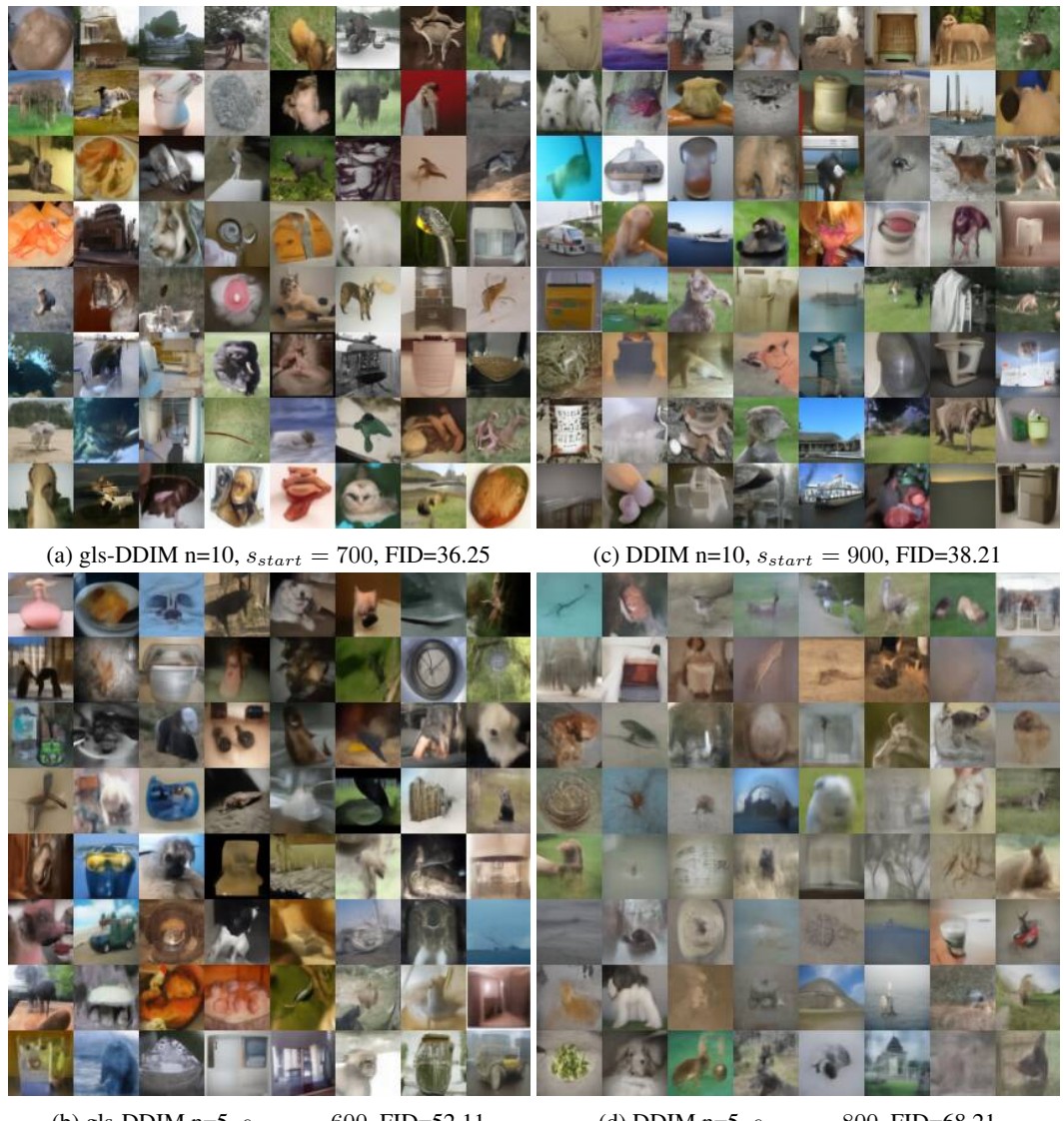

(a) gls-DDIM n=10, $s_{start} = 700$, FID=36.25      (c) DDIM n=10, $s_{start} = 900$, FID=38.21

(b) gls-DDIM n=5, $s_{start} = 600$, FID=52.11      (d) DDIM n=5, $s_{start} = 800$, FID=68.21

Figure 24: Our Gaussian late start initialization boost performance on DDIM for 10 and 5 denoising steps on Imagenet64.

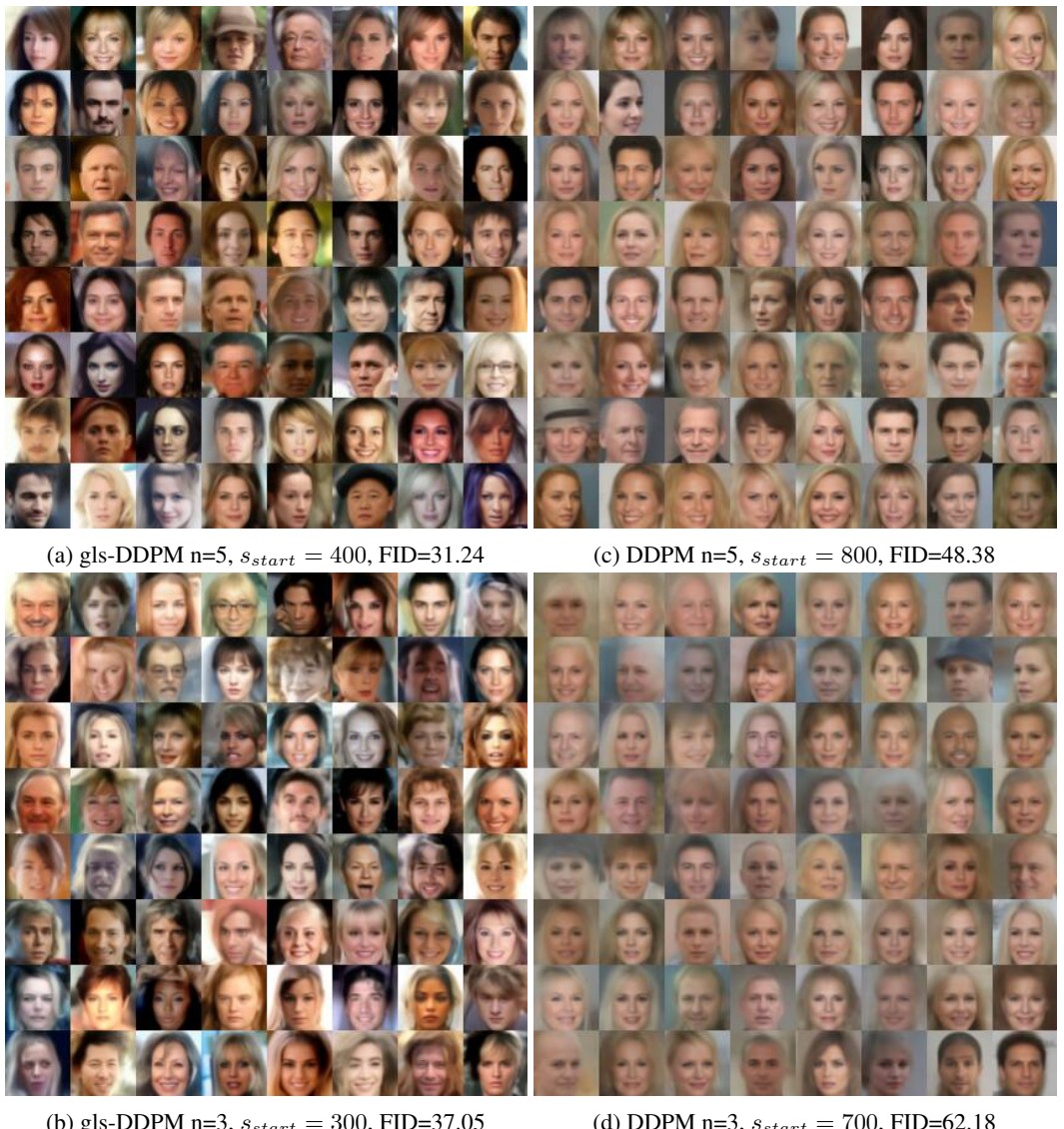

(a) gls-DDPM n=5, $s_{start} = 400$, FID=31.24

(c) DDPM n=5, $s_{start} = 800$, FID=48.38

(b) gls-DDPM n=3, $s_{start} = 300$, FID=37.05

(d) DDPM n=3, $s_{start} = 700$, FID=62.18

Figure 25: Our Gaussian late start initialization boost performance on DDPM for 5 and 3 denoising steps on Celeba64.

# D Diversity analysis

This section presents an expanded examination of the diversity analysis carried out on CelebA64 samples generated using the DDIM sampler, and our gls-DDIM initialization. Figure 26 illustrates the analysis of "emotion"" and "gender" attributes for 5 denoising steps. Meanwhile, Figure 27 provides a visual summary of the "age" distribution for 5 and 10 denoising steps. For comparison, we also provided the diversity analysis obtained in the training set and from the standard DDPM sampler with 1000 denoising steps (DDPM-1000).

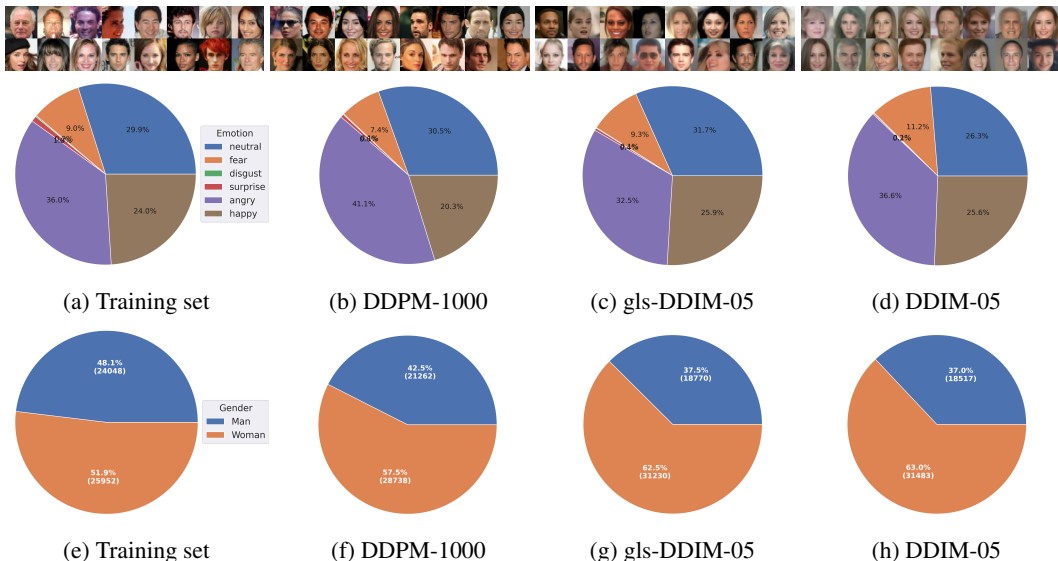

Figure 26: "Emotion" and "Gender" diversity analysis on CelebA64 over 50,000 generated samples by (c,g) gls-DDIM and (d,h) DDIM samplers with 5 denoising steps. Results obtained on (a,e) training set and (b,f) DDPM using 1000 denoising steps are provided for reference. Corresponding samples obtained by each set are shown on top of the pie charts.

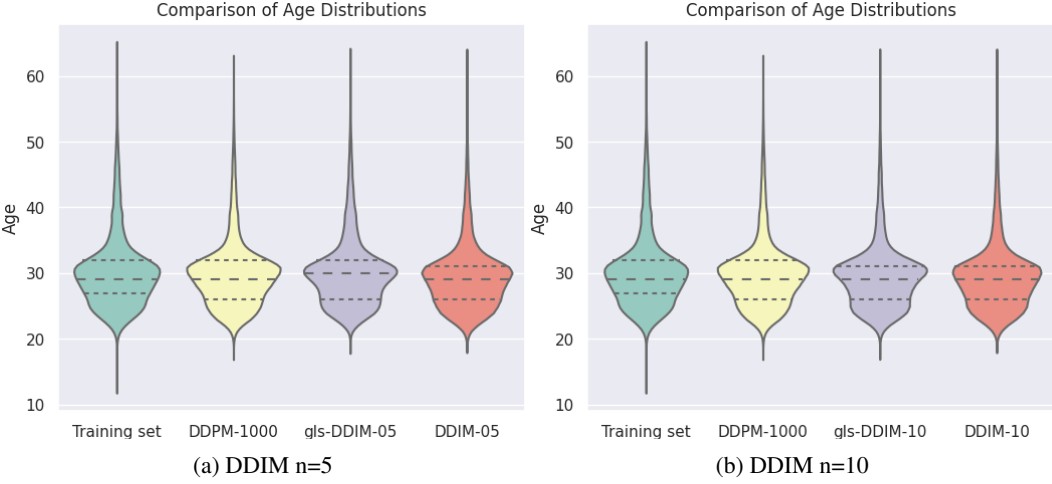

Figure 27: Age attribute analysis on generated CelebA64 samples for 5 and 10 denoising steps.

# E Implementation Details

Our implementation is based on a newly developed codebase, taking inspiration from the implementation by Song et al. Song et al. (2021) for the DDPM model. In our experiments, we employ DDPM models where the stochastic differential equation (SDE) is defined over the continuous-time interval $t \in [0, 1]$ and discretized into $N = 1000$ time steps, representing a finite horizon of $T = 1$ in discrete-time. We conducted all experiments on NVIDIA DGX-1 machines with 3 Tesla V100 GPUs each, utilizing PyTorch 1.10.2+cu102, CUDA 10.2, and CuDNN 7605.

## E.1 FID computation

To compute FID scores, we use the Inception-v3 model to extract activations from the coding layer for both real and generated images. We calculate the mean and covariance matrix of these activations for the training set and over 50,000 generated images separately. We validate our implementation by comparing FID scores with previous work Ho et al. (2020); Song et al. (2021), such as achieving a FID score of 3.08 on CIFAR-10.

## E.2 Implementation of the one-dimensional diffusion model

Following Song et al. (2021), the implementation of the time-continuous function $\beta(s)$, with $s = T - t$ and $s \in [0, 1]$, is given by $\beta(s) = \bar{\beta}_{min} + s(\bar{\beta}_{max} - \bar{\beta}_{min})$ discretized over $N = 1000$ steps. To match our DDPM settings in realistic datasets, we let $\bar{\beta}_{min} = 0.1$ and $\bar{\beta}_{max} = 20$. Therefore we can estimate the evolution of $\theta_s = e^{-\frac{1}{2} \int_0^s \beta(\tau)d\tau}$ as follows:

$$
\begin{aligned}
\theta_s &= e^{-\frac{1}{2} \int_0^s \beta(\tau)d\tau} \\
&= e^{-\frac{1}{2} \int_0^s \bar{\beta}_{min} + \tau(\bar{\beta}_{max} - \bar{\beta}_{min})d\tau} \\
&= e^{-\frac{1}{4} s^2 (\bar{\beta}_{max} - \bar{\beta}_{min}) - \frac{1}{2} s \bar{\beta}_{min}}
\end{aligned}
$$

## E.3 Code Availability

Our code can be found at https://github.com/gabrielraya/symmetry_breaking_diffusion_models