# OpenReview forum: "Spontaneous symmetry breaking in generative diffusion models"
_NeurIPS.cc/2023/Conference — NeurIPS 2023 poster_

### Official Review · Reviewer_Z6e7 · 2023-07-05

**Soundness:** 3 good
**Presentation:** 3 good
**Contribution:** 3 good
**Rating:** 5
**Confidence:** 4

**Summary:**

The paper explores the generative dynamic of diffusion models. It proposes spontaneous symmetry breaking and Gaussian late initialization scheme and they achieve better fidelity and diversity on the generated images.

**Strengths:**

[S1] The paper supports the claims in both theoretical and empirical.

[S2] The paper successfully convince the proposed approach improves diversity by comparing.

**Weaknesses:**

[W1] They do not discuss the limitations and drawbacks of their model.

[W2] The other generative models such as VAE, GAN, regression models, etc. do not discuss. At least, it is expected to mention them in the related work section as they claim that the proposed approach has a broader impact on generative models.

**Questions:**

[Q1] In Table 1, why the dataset and denoising step (n) is repeated? You can put them in the first column and it might be enough. So, instead of making subtables, can you make them a single table?

[Q2] In Table 1 and Figure 7 why you didn't include other generative models such as GANs in comparison?

[Q3] Is Table 1 an ablation study to show the effect of the gls? If so, where is the fidelity comparison of the proposed approach?

[Q4] In the discussion of Table 1, you do not discuss the results so as to indicate the spontaneous symmetry breaking. What is the reason for this?

**Limitations:**

[L1] It is only applicable to the diffusion models.

[L2] The paper ignores the other type of generative models in both related work and experiments. It is required to compare the proposed approach with existing generative models.

[L3] The paper does not discuss the drawback of the proposed approach.

---

> ### Author Rebuttal · Authors · 2023-08-08
>
> We appreciate the reviewers' feedback and are happy to hear that our work supports our claims both theoretically and empirically. In the following, we will address the main issues raised by the reviewer.
>
> ### Point 1: Discussion of other generative models
> Q: *“The other generative models such as VAE, GAN, regression models, etc. do not discuss. At least, it is expected to mention them in the related work section as they claim that the proposed approach has a broader impact on generative models.”*
>
> A: In the revised version, we will add a sentence discussing other generative models in the introduction and a more extensive discussion about how to generalize these insights to other models in the discussion section.
>
> We indeed focused our attention on generative diffusion models. This is due to the fact that these models are radically different from other generative approaches such as VAEs and GANs and it is not straightforward to extend these insights to these other models. However, given the current importance of diffusion models in the generative modeling literature, we believe that the insights developed by carefully analyzing these models can have a broad impact on generative modeling.
>
> ### Point 2: Experimental comparison with other generative models
> Q: *“In Table 1 and Figure 7 why you didn't include other generative models such as GANs in comparison?”*
>
> A: Our aim is 1) to demonstrate that spontaneous symmetry breaking phenomena are ubiquitous in the dynamics of generative diffusion models and 2) to show that this insight can lead to faster, higher quality and more diverse sampling in diffusion models.
>
> Given these aims, it is unclear what we would learn by including other non-diffusion baselines. The relative performance of these models when compared with diffusion models has already been discussed in the literature [1, 2] and it is orthogonal to our claims.
>
> [1] Diffusion Models Beat GANs on Image Synthesis, https://arxiv.org/abs/2105.05233
> [2] Improved Denoising Diffusion Probabilistic Models, https://arxiv.org/abs/2102.09672
>
>
> ### Point 3: Table 1
> Q: *“In the discussion of Table 1, you do not discuss the results so as to indicate the spontaneous symmetry breaking. What is the reason for this?”*
>
> A: Unfortunately, we are not sure to have understood the question. We show the existence of the symmetry breaking in the FID curves in Fig. 4 and in the splitting of the potential obtained from the model in Fig.1 (right side) and Supps in Fig. 13-14. Table 1 reports the result of the late initialization experiment. As we explain in the paper, the existence of the symmetry breaking predicts that late initialization will produce more efficient fast samples, which is confirmed in Table 1. We hope this clarifies it.

---

> > ### Comment · Reviewer_Z6e7 · 2023-08-16
> > **Response to Rebuttal**
> >
> > Thank you for the rebuttal. I think it is good to enlarge the discussion by adding other generative (types of) models to demonstrate a complete picture and allow others to have better insight into how the idea can be generalized. Besides, I carefully read the rebuttal and the other reviews and rebuttals and reconsidered my rating as borderline accept.

---

### Official Review · Reviewer_FFjV · 2023-07-07

**Soundness:** 2 fair
**Presentation:** 3 good
**Contribution:** 3 good
**Rating:** 4
**Confidence:** 4

**Summary:**

The authors approach the reverse diffusion process in score-based generative modeling from the perspective of spontaneous symmetry breaking, which may explain why generation is minimally affected by "late initializations" (e.g. initializing x_t at t=T, but simulating the reverse process starting at some t < T). Motivated by this framework, the authors propose late-start scheme and show improved performance compared to some baselines.

**Strengths:**

**Exciting direction of study.** The authors propose an exciting perspective on the diffusion model in terms of spontaneous symmetry breaking in physical systems.

**Initially promising empirical results.** The empirical results appear to be promising. (However, they do not presently provide much in the way of concrete proof that the symmetry breaking framework provides tangible improvements on generation speed or quality, see "Weaknesses").

**Weaknesses:**

**Motivation.** The motivation of the work should be made more clear. If it is to improve generation speed, the model should be compared against fast generation models, such as Model Distillation, Consistency Models, and EDM. If it is to generate better samples, then the model should be compared against standard diffusion models such as [4] or [5].

**Empirical Results.** The empirical results are not very convincing. All models achieve FIDs that are far from state-of-the-art. It is not clear at this stage whether improvements come directly from the proposed approach, and whether the improvements will remain when models are tuned to be more competitive with respect to the state of the art.

**Claims.** The claim (up to 3x FID improvements) feels overstated. The authors do not compare against state-of-the-art fast samplers, such as Model Distillation [1], Consistency Models [2], and EDM [3].

**Soundness.** I have some concerns over the soundness of the claims in the paper.
- The existence of stable points is only shown for toy examples (e.g. Secs. 3.1 and 3.2). Even in these toy examples, it is not clear to me how the stable points will affect the result of the reverse diffusion process for most initial conditions of the reverse process. Showing the existence of stable points at the origin and the original data points implies only the existence of a finite set of stable points, which are essentially probability zero set of the entire space. Diffusion models are initialized from a standard normal, so x_T can be very far from the origin (or any of the stable points). So is it true that the spontaneous symmetry breaking model is relevant here, even in theory?
- While the second toy example is more realistic (Sec 3.2), and the authors claim that the assumptions required are mild, I do not believe they hold for any usual image distributions (e.g. MNIST, CIFAR10, ImageNet, CelebFaces, etc.).
- Finally, the authors connect the theory of symmetry breaking to diffusion modeling with a single piece of empirical evidence: The FID of samples drawn with late start diffusion is relatively stable (up to a critical) point in time t_c. I can see how this is necessary condition to show the existence of symmetry breaking. But how is it sufficient?

In summary, while the theory of spontaneous symmetry breaking is very interesting and thought-provoking, I am not yet convinced that it is presently a useful perspective in diffusion-based generative modeling.

**Minor edits.**

39: the theory weak and electromagnetic -> the theory of weak and electromagnetic

108: Hessian matrices (of what?)

[1] Progressive Distillation for Fast Sampling of Diffusion Models. https://arxiv.org/abs/2202.00512

[2] Consistency Models. https://arxiv.org/abs/2303.01469

[3] Elucidating the Design Space of Diffusion-Based Generative Models. https://arxiv.org/abs/2206.00364

[4] Denoising Diffusion Probabilistic Models. https://arxiv.org/abs/2006.11239

[5] Score-Based Generative Modeling through Stochastic Differential Equations.https://arxiv.org/abs/2011.13456

**Questions:**

See "Weaknesses" section.

**Limitations:**

See "Weaknesses" section.

---

> ### Author Rebuttal · Authors · 2023-08-08
>
> We are thankful to the reviewer for the comprehensive and insightful feedback. In what follows, we'll address the main concerns and questions.
>
> ### Point 1: Motivation
> Q: *“The motivation of the work should be made more clear. If it is to improve generation speed, the model should be compared against fast generation models, such as Model Distillation, Consistency Models, and EDM. If it is to generate better samples, then the model should be compared against standard diffusion models such as [4] or [5].”*
>
> A: We acknowledge that we were not entirely clear in the original manuscript and we will change the text accordingly. Our primary goal is to demonstrate that spontaneous symmetry breaking is a near-universal consequence of the dynamics of generative diffusion models. Our result establishes a direct connection between a dominant class of generative models in machine learning and a large body of theory and numerical methodologies developed in physics, which can be leveraged to improve our understanding and mastery of generative AI.
>
> As we argue in the general reply to all reviewers, the goal of our empirical study on fast samplers is to show how this theoretical understanding can be leveraged in practice and it can indeed lead to rather impressive results. However, we believe that the level of optimization needed to optimize SOTA methods goes beyond the scientific scope of this paper.
>
>
> ### Point 2: Comparison with distillation models
> Q: *“the model should be compared against fast generation models, such as Model Distillation, Consistency Models, and EDM”*
>
> A: Distillation models aim at skipping the generative dynamics in order to obtain one or few step sampling, often using a re-trained model. Since our aim is to study the generative dynamics, we decided to not focus on this class of methods. So said, it is worth noting that distillation models still rely on the DDIM sampler in their training process, a sampler that we have employed and improved upon in our paper.
>
> ### Point 3: Relevance of the fixed-points
> Q: *“The existence of stable points is only shown for toy examples (e.g. Secs. 3.1 and 3.2). Even in these toy examples, it is not clear to me how the stable points will affect the result of the reverse diffusion process for most initial conditions of the reverse process. Showing the existence of stable points at the origin and the original data points implies only the existence of a finite set of stable points, which are essentially probability zero set of the entire space. Diffusion models are initialized from a standard normal, so x_T can be very far from the origin (or any of the stable points). So is it true that the spontaneous symmetry breaking model is relevant here, even in theory?”*
>
> A: This is a very insightful and subtle point. Indeed, the reviewer is entirely correct in pointing out that, at least in the early stage, the fixed-point in itself does not seem to be very relevant. What actually matters is the shape of the potential around the fixed-point. The main reason to study the bifurcation in the fixed-points is that it implies a dramatic change of shape in the potential, which splits into multiple wells, each representing a subset of the original symmetry group. This change of shape is the hallmark of a spontaneous symmetry breaking, as visualized in Figure 2 in the attached pdf.
>
> As a secondary point, note that for  $T-t <T- t_c$ the fixed-points are not always isolated. In fact, if the data spans a d-dimensional manifold with uniform probability, for $T-t$ tending to $0$, each point in the manifold becomes a fixed-point. Finally, in a discrete dataset, for $T-t$ tending to zero, the fixed-points will generally acquire a non-zero measure since the magnitude of the score pointing at them tends to infinity.
>
> ### Point 4: Assumptions in discrete datasets
> Q: *“While the second toy example is more realistic (Sec 3.2), and the authors claim that the assumptions required are mild, I do not believe they hold for any usual image distributions (e.g. MNIST, CIFAR10, ImageNet, CelebFaces, etc.).”*
>
>
> A: We would like to point out that we did not claim that these assumptions are met in real datasets, but instead that it is straightforward to induce them (at least approximately) by normalization. For example, subtracting the mean (centering) is commonly done as a preparatory step in many algorithms without loss of generality, since the mean can be summed after generating the centered data. In general, the centering assumption is only needed to make sure that the fixed-point is the origin and to make the analysis simpler. Concerning the constraint on the Euclidean norm, it is well known that the Euclidean norm often concentrates around a single value in high-dimension, meaning that the data approximately ‘live’ in a hyper-spherical annulus. Generally speaking, while the constraint is not exactly met, we think that it is relevant for understanding high-dimensional data.
>
>
> ### Point 5: Empirical evidence
> Q: *“Finally, the authors connect the theory of symmetry breaking to diffusion modeling with a single piece of empirical evidence: The FID of samples drawn with late start diffusion is relatively stable (up to a critical) point in time t_c. I can see how this is necessary condition to show the existence of symmetry breaking. But how is it sufficient?”*
>
> A: The late start FID curves are not the only empirical result in the paper. Using the trained network, we also directly evaluate the potential on arcs connecting two trajectories and we show that the potential has a single minimum initially and then splits into two separate minima. This can be seen in Figure 1 and in the Supplementary Material in Figures 13-15. These changes in the shape of the potential provide very direct empirical evidence of the symmetry breaking phenomenon in trained models.

---

### Official Review · Reviewer_kBdj · 2023-07-09

**Soundness:** 4 excellent
**Presentation:** 3 good
**Contribution:** 3 good
**Rating:** 6
**Confidence:** 4

**Summary:**

The authors investigate theoretically and empirical when (in terms of diffusion timestep) certain symmetries in a diffusion process are broken, corresponding to when "choices" are made about which qualitative features a generated data point should have. Their findings suggest that there is no symmetry breaking occuring early in the process. They follow this up by showing that the early dynamics can be replaced by a sample from a multivariate Gaussian without meaningful degradation of the perceptual quality of generated data.

**Strengths:**

- This perspective on the diffusion process is novel as far as I know.
- Understanding when in the diffusion process different symmetries are broken is likeliy to be very helpful for future work designing better diffusion-based models.
- Their technique for better fast sampling yields surprising large improvements. Although they acknowledge that it does not immediately scale to high-dimensional data, it provides good support of their analysis.

**Weaknesses:**

- There is a fairly large gap between the simple example in Section 3.1, where we could reason analytically about when the symmetry is broken, to the realistic case in Section 3.2, where the only symmetries considered were the identity of each data point. Do the authors think it would be possible or meaningful to do this type of analysis on other features that could be extracted from the data (e.g. the identity of a person in an image)?
- This analysis of symmetry breaking makes sense for "discrete" variables, like cluster index. It does not significantly help to provide insights about when values of "continuous variables" like e.g. background colour of an image are selected.
- The proposed Gaussian initialization improves fast samplers in terms of FID and the authors show is helpful in terms of maintaining the identity of discrete variables, but it is important to note that replacing the early diffusion process with a Gaussian may still lead to large changes in the distribution.

**Questions:**

See weaknesses (mainly the first one)

**Limitations:**

sufficiently addressed

---

> ### Author Rebuttal · Authors · 2023-08-08
>
> We thank the reviewer for the valuable feedback. Next, we will address the highlighted concerns and questions.
>
> ### Point 1: Gap between theory and experiments
> Q: *“There is a fairly large gap between the simple example in Section 3.1, where we could reason analytically about when the symmetry is broken, to the realistic case in Section 3.2, where the only symmetries considered were the identity of each data point. Do the authors think it would be possible or meaningful to do this type of analysis on other features that could be extracted from the data (e.g. the identity of a person in an image)?”*
>
> A: There is indeed a substantial gap between what we can show theoretically in analytically tractable models with simple symmetry groups and the actual (often approximate) symmetries in real-world datasets. We definitely agree that a substantial amount of work can be done in characterizing the symmetry breaking for more complex symmetry groups (e.g. translational and rotational symmetries) and also, as the reviewer suggested, on symmetry transformation learned directly from a dataset. However, these are significant and complex endeavors that go beyond the scope of this first paper. While we understand the perspective that this might be seen as a weakness, in fact, we do believe this is a significant strength of our paper, as it opens several fascinating research directions.
>
> ### Point 2: Discrete and continuous symmetries
> Q: *“This analysis of symmetry breaking makes sense for "discrete" variables, like cluster index. It does not significantly help to provide insights about when values of "continuous variables" like e.g. background colour of an image are selected.”*
>
> A: In the main text, we focused on discrete symmetry groups as they show the phenomenon in a clear and simple way. However, our symmetry breaking framework can deal with both continuous and discrete symmetry groups. In fact, we discussed the case of the continuous $SO(N)$ symmetry breaking in the Supplementary Material in section A.3. In order to make this point clear, we will move this analysis to the main text. We will also include a new paragraph discussing this point explicitly.
>
> ### Point 3: Deviation from Gaussianity
> Q: *"The proposed Gaussian initialization improves fast samplers in terms of FID and the authors show is helpful in terms of maintaining the identity of discrete variables, but it is important to note that replacing the early diffusion process with a Gaussian may still lead to large changes in the distribution".*
>
> A: While it is true that the distribution will not be exactly Gaussian, our theoretical and, more importantly, experimental results show that the deviations are minor during the first phase.
> To further address this point, we performed Gaussianity tests on forward samples as a function of time. Due to Anderson's theorem, both forward and generated samples have the same marginal statistics. The results of these tests across several datasets are graphically illustrated in Figure 1 on the attached pdf.

---

> > ### Comment · Reviewer_kBdj · 2023-08-17
> >
> > Thanks for the response, which addressed my concerns. I will keep my rating of 6.

---

### Official Review · Reviewer_J7iR · 2023-07-10

**Soundness:** 3 good
**Presentation:** 3 good
**Contribution:** 3 good
**Rating:** 7
**Confidence:** 4

**Summary:**

The authors of this work propose a new method to accelerate the sampling of diffusion models. First, the authors define as fixed points of the reverse process, points where the drift function is 0. Then, the authors claim that if two paths of stable fixed points interact, then it is the noise that can make the system jump from one path to the other. The authors argue that there are phase transitions happening in the sampling of diffusion models: until a certain time, the stable paths interact and hence the system jumps between potential paths. After a point in the reverse diffusion, there are no more intersections in the paths. Even if we don't initialize exactly in the path, the stability ensures that the differential equations are mean-reverting and will correct any introduced errors. Hence, the authors propose to start the sampling from a point in the middle of the diffusion and accelerate the sampling.

**Strengths:**

The introduced framework is very interesting and novel. The example with the two Dirac functions is very pedagogical and its' generalization to a uniform distribution over many discrete points follows naturally. The experimental evidence supports that this phenomenon is indeed happening in trained diffusion models. The proposed method surpasses the baselines for a low number of function evaluations. The paper is well-written and the approach can be implemented easily. I expect that the paper will be of great interest to the research community and the audience of NeurIPS.

**Weaknesses:**

The theoretical analysis has some simplifying assumptions that it is not clear to what extent they alter what happens in practice. Specifically, the authors analyze uniform distributions over Dirac functions. The distributions observed in practice have a more complicated structure and it is not clear how we can argue anything about the fixed points of a model given access to its (learned) score-function.

A weakness of the approach is that to start the sampling from some intermediate diffusion time, one needs to estimate the parameters of a multivariate Gaussian. This can be a particularly challenging task and it doesn't scale well for high dimensions because of the quadratic parameters needed for the covariance. The authors acknowledge this limitation.

Experimentally, the authors mostly consider what happens in the low NFEs regime. More detailed comparisons would be useful. If the method performs poorly (relative to the baselines) for higher NFEs, it is important to acknowledge it.

**Questions:**

* It seems to me that the method can be used *together* with other approaches for accelerating sampling (since it is only changing the initial point). It would strengthen the paper if the authors included experiments to show this.
* Can the authors comment on whether we can find fixed points given a trained diffusion model?
* I am puzzled by what happens to the story that this paper builds once we think about the deterministic samplers. In such settings, stable points would correspond to no movement at all. If the system becomes mean-reverting at some point, wouldn't that mean that we stop moving once we reach a fixed point?
* It would strengthen the paper to include comparisons with EDM and other more recent sampling methods.
* It would also help to include results for higher NFEs.

**Limitations:**

The authors adequately addressed the limitations.

---

> ### Author Rebuttal · Authors · 2023-08-08
>
> We wish to thank the reviewer for the detailed and insightful review. In what follows, we will address the main issues and questions.
>
> ### Point 1: Analytically tractable models
> Q: *“The theoretical analysis has some simplifying assumptions that it is not clear to what extent they alter what happens in practice. Specifically, the authors analyze uniform distributions over Dirac functions.”*
>
> A: The Z1 symmetry breaking on the on-dimensional model has been chosen due to its pedagogical value and the fact that it highlights all the main conceptual points. However, this form of binary symmetry breaking has relevance in many situations, for example, if the data spans two topologically separated manifolds. Of course, in the real model the Z1 symmetry will be a small subgroup of the overall symmetry group.
>
> ### Point 2: Studying the fixed-point in trained models
>  Q: *"The distributions observed in practice have a more complicated structure and it is not clear how we can argue anything about the fixed points of a model given access to its (learned) score-function.” “Can the authors comment on whether we can find fixed points given a trained diffusion model?”*
>
> A: In theory, having access to a fully learned score function allows us to analyze its fixed points, which can be found by gradient descent. Specifically, this could be used to find the original fixed point, to determine its stability by evaluating the Hessian, and consequently to detect the onset of a symmetry breaking in a real-world model. However, unfortunately, this analysis is not going to be reliable in actual trained models since the fixed-point itself is outside the training range, and the shape of the potential in that region is not properly trained. This is due to the, perhaps paradoxical, fact that the fixed points have vanishingly low probability of being visited, as the samples concentrate in a fixed-variance annulus around them (see reply to Reviewer FFjV, Point 3). However, what ultimately matters is the shape of the potential, which we have studied directly in trained models along variance-preserving arcs connecting pairs of data points, where we found the expected change of shape (see Figure 1 and in the Supplementary Material see Figures 13-15).
>
> ### Point 3: Scalability of the Gaussian late initialization
> Q: *"A weakness of the approach is that to start the sampling from some intermediate diffusion time, one needs to estimate the parameters of a multivariate Gaussian. This can be a particularly challenging task and it doesn't scale well for high dimensions because of the quadratic parameters needed for the covariance. The authors acknowledge this limitation."*
>
> A: Indeed, in fact the GLS as used in the paper is meant as a proof of principle to showcase the importance of an appropriate initialization in order to preserve diversity. So said, it is rather straightforward to make the approach scalable either by PCA, Fourier analysis or, even better, by replacing the Gaussian initialization with another more efficient model such as a VAE.
>
> ### Point 4: Performance for a large number of samples
> Q: *“Experimentally, the authors mostly consider what happens in the low NFEs regime. More detailed comparisons would be useful. If the method performs poorly (relative to the baselines) for higher NFEs, it is important to acknowledge it.”*  *“It would also help to include results for higher NFEs.”*
>
> A: Certainly, it is important to present results on higher NFEs. Thank you for pointing that out. In light of your feedback, we have incorporated extra experiments for 20, 50, and 100 denoising steps. You can view these results in Table 1 on the attached PDF. Notably, we observe the same behavior obtained at low NFEs remains at higher NFEs. Consequently, our gls scheme enhances performance over fast samplers, like DDIM and PNDM, as well as the standard DDPM.
>
> ### Point 5: Deterministic samplers
> Q: *“I am puzzled by what happens to the story that this paper builds once we think about the deterministic samplers. In such settings, stable points would correspond to no movement at all. If the system becomes mean-reverting at some point, wouldn't that mean that we stop moving once we reach a fixed point?”*
>
> A: This is a very interesting point. Since the deterministic samplers are designed to exactly track the marginals of the stochastic sampler, a change in the stochastic dynamics will also appear in the deterministic case. However, the symmetry breaking looks very different from the point of view of the deterministic samplers. Specifically, in the deterministic ODEs, the correction term makes the drift vanish prior to the symmetry breaking. Therefore, in a deterministic sampler the first phase is ‘irrelevant’ not because of the mean-reverting dynamics, but just because there is no significant temporal evolution prior to that point. Therefore, with good approximation every point is a fixed-point in the first phase of the deterministic ODE. This can be clearly seen in Fig. 7(k) and Fig 16 (c,f,i,l,o)  in the Supplementary Material.

---

> > ### Comment · Reviewer_J7iR · 2023-08-12
> > **Response to Rebuttal**
> >
> > Thank you for the rebuttal. I will keep my score which recommends acceptance. Please incorporate the rebuttal discussion in the main paper, if possible.

---

### Official Review · Reviewer_sN4H · 2023-07-14

**Soundness:** 4 excellent
**Presentation:** 4 excellent
**Contribution:** 3 good
**Rating:** 7
**Confidence:** 3

**Summary:**

The authors examine denoising diffusion models and their dynamics through the lense of spontaneous symmetry breaking.  They characterize the dynamics of the denoising process into two stages: in the beginning, samples orbit a central fixed point and after a critical amount of time denoising of a ‘selected’ sample begins.  They use insights gained from theory as well as practical examples to develop a novel fast sampling method they call “Gaussian late start” (GLS).


**Strengths:**

This was a very well written paper.  The authors motivate the issue very well, and they additionally illucidate the concept of spontaneous symmetry breaking in a way that is easy to understand.

One of the biggest strengths, was the foundational example laid out in Sec. 3.1 — this example helped fill in a lot of gaps I had up until that point, and made me really appreciate the authors characterization of the denoising dynamics into two separate stages.

The experiment of section 4.1 followed by its extension in section 4.2 do appear to give credence to thinking of the denoising dynamics as having a spontaneous symmetry breaking characteristic.  This also leads in nicely to the Gaussian late start sampling method, which is both intuitive, practical, and appears to work well.

Overall, the paper presents a compelling idea, motivates it through practical toy examples as well as theory, and then provide a novel efficient way of sampling.

**Weaknesses:**

It seems like this is not a problem constrained solely to diffusion models used for image generation, but for any data modality.  In that case, it might be good to empirically demonstrate  this for another data modality.

**Questions:**

This may be a silly question, but I was wondering if the authors thought if the datasets clustered and the sampler was initialized using the statistics of each cluster — do you expect the samples drawn to be similar to the cluster whose statistics were used to initialize the sampler?

**Limitations:**

As noted, GLS as is, is slightly impractical, but there is room for future research to develop fast approximate methods in the same vain.

---

> ### Author Rebuttal · Authors · 2023-08-08
>
> We wish to thank the reviewer for the positive and insightful review. We are happy to hear that you found the one-dimensional example as insightful, as it certainly shaped our understanding of this fascinating phenomenon. To further solidify the intuition, in the updated manuscript we will also add a figure visualizing the fixed-points and showing how the score changes after the bifurcation (see Figure 2 in the attached pdf).
>
> In the following, we will address some of your questions.
>
>
> ### Point 1: Other modalities
> Q: *“It seems like this is not a problem constrained solely to diffusion models used for image generation, but for any data modality. In that case, it might be good to empirically demonstrate this for another data modality.”*
>
> A: We agree that the phenomenon should not be limited to image datasets. The choice of the experiments mainly reflected our field of expertise. We also agree that it would be useful to see results in other modality, such as audio and language, and we are planning to pursue these research  directions in the future.
> ### Point 2: Cluster initialization
> Q: *“This may be a silly question, but I was wondering if the authors thought if the datasets clustered and the sampler was initialized using the statistics of each cluster — do you expect the samples drawn to be similar to the cluster whose statistics were used to initialize the sampler?”*
>
> A: Interesting point. Prior to the symmetry breaking, all clusters are merged into a single mode (around a fixed-point). Therefore, at least if we are talking about stochastic samplers, the initial value will not correlate with the final cluster assignments. The situation is different for deterministic samplers. In this case, the initialization scheme will likely work. However, we believe that it risks inducing significant overfitting.

---

> ### Comment · Reviewer_sN4H · 2023-08-17
>
> Thank you to the authors for answering my questions. I maintain my original score which is to accept.

---

### Author Rebuttal · Authors · 2023-08-08

Dear reviewers,

We wish to thank you for the detailed analysis of our paper and for all the encouraging and constructive comments. We are extremely grateful for your work, as it will make our manuscript substantially better. Several important questions were raised, and we provided extensive answers under the individual reviews. In particular, we would like to point out to the discussion in the reply to reviewer  FFjV (point 3) about the importance of the fixed points and the analysis of the deterministic samplers in the reply to reviewer J7iR (point 5). Upon acceptance, we will incorporate these discussions in the updated manuscript.

In what follows, we will outline our conclusions from the reviews, addressing both the strengths and weaknesses of our work as pointed out by the reviewers. We have also included a summary of all the additional analysis that we conducted during the rebuttal period in response to questions and observations raised by the reviewers.

### **Main strengths**
We are happy to see that all reviewers agreed that our main claim is well supported by strong theoretical and empirical evidence. This is corroborated by the fact that two reviewers (sN4H and kBdj) marked the soundness of the paper as excellent, a distinction that makes us proud, given the ambitious nature of our claims. We are also pleased to see that a majority of reviewers agree that our work is novel and important, as expressed, for example, by the following quotes from reviewer J7iR and kBdj:

> *“I expect that the paper will be of great interest to the research community and the audience of  NeurIPS.”*  (reviewer J7iR) ,

> *“Understanding when in the diffusion process different symmetries are broken is likely to be very helpful for future work designing better diffusion-based models.”* (reviewer kBdj) ,

and even by the more critical reviewer FFjV

> *“the theory of spontaneous symmetry breaking is very interesting and thought-provoking”* (reviewer FFjV) .


### **Main weaknesses**
The main concern, most prominently expressed by reviewer FFjV, is that the improvements in sample quality and diversity due to the Gaussian late initialization scheme might not generalize to state-of-the-art (SOTA) fast-sampling approaches such as progressive distillation and consistency models.

While we acknowledge that this is a fair criticism, we would like to emphasize that the primary motivation of this paper is not to propose a SOTA fast sampler. The primary goal of the paper is instead to demonstrate that the generative dynamics of diffusion models can be understood as spontaneous symmetry breaking, a phenomenon that is ubiquitous  in physical systems.  This is in itself a very ambitious endeavor as it establishes a direct connection between generative AI and some of the deepest aspects of fundamental physics (e.g. the standard model and the statistical mechanics of critical phenomena), which might in turn lead to substation methodological and theoretical developments in both fields.

The purpose of our sampling experiments is indeed to demonstrate that understanding this phenomenon can directly lead to algorithmic improvements, thereby providing evidence for its usefulness.  However, reaching the boundaries of the state-of-the-art requires a level of optimization that we believe goes beyond the scope of our paper. In this sense, our sampling experiments should be interpreted as a proof of principle and a starting point for future developments.

### **Summary of new experimental results**
Several questions and observations raised by the reviewers motivated us to run several new experiments for this rebuttal period, which will end up in the revised version of the manuscript.

In the accompanying PDF, we provide expanded analyses addressing our reviewers' questions.

- Reviewer kBdj concern about the Gaussianity nature of the approximate distribution is addressed in Figure 1, which provides a Gaussianity analysis via the Shappiro-Wilk test. The figures corroborate the validity of a Gaussian distribution up until the symmetry breaking’s critical point, beyond which the distribution rapidly ceases to be Gaussian.
- In answer to the comments made by reviewer J7iR, Table 1 expands on the robustness of our improvements in the context of higher numbers of function evaluations (NFE). Here, we present results for 20, 50, and 100 denoising steps for datasets including MNIST, CIFAR10, and Celeba64. These results confirm that our improvements persist in various NFE regimes.
- Lastly, to address reviewer FFjV concern on the importance of the fixed-points, Figure 2 provides a visualization of the bifurcation of the fixed-points (estimated numerically) in the one-dimensional model, together with a visualization of the vector field provided by the score function. The figure shows how the bifurcation is associated with a dramatic shift in the vector field, which determines the dynamics of the system away from the fixed-points themselves.

---

> ### Author Response · Authors · 2023-08-21
>
> Dear Reviewers,
>
> We would like to thank you once again for your insightful reviews. We are encouraged by your positive and constructive feedback, which will lead to an improved manuscript.

---

### Decision · Program_Chairs · 2023-09-21

**Decision:**

Accept (poster)

**Comment:**

Most reviewers appreciate the scientific value of revealing a nature in the working mechanism of diffusion model under the view of a dynamical system, and the practical value of a derived faster sampling implementation. There are also concerns on the assumptions for the analysis, and practicality of the proposed method. The ratings indicate the contributions overweigh the concerns. I hence recommend accept for this submission.